# Neural Learning of One-of-Many Solutions for Combinatorial Problems in Structured Output Spaces

**Yatin Nandwani\*, Deepanshu Jindal**[*]**, Mausam & Parag Singla**
Department of Computer Science, Indian Institute of Technology Delhi, INDIA
`{yatin.nandwani, deepanshu.jindal.cs116, mausam, parags}@cse.iitd.ac.in`

## Abstract

Recent research has proposed neural architectures for solving combinatorial problems in structured output spaces. In many such problems, there may exist multiple solutions for a given input, e.g. a partially filled Sudoku puzzle may have many completions satisfying all constraints. Further, we are often interested in finding *any one* of the possible solutions, without any preference between them. Existing approaches completely ignore this solution multiplicity. In this paper, we argue that being oblivious to the presence of multiple solutions can severely hamper their training ability. Our contribution is two fold. First, we formally define the task of learning one-of-many solutions for combinatorial problems in structured output spaces, which is applicable for solving several problems of interest such as N-Queens, and Sudoku. Second, we present a generic learning framework that adapts an existing prediction network for a combinatorial problem to handle solution multiplicity. Our framework uses a selection module, whose goal is to dynamically determine, for every input, the solution that is most effective for training the network parameters in any given learning iteration. We propose an RL based approach to jointly train the selection module with the prediction network. Experiments on three different domains, and using two different prediction networks, demonstrate that our framework significantly improves the accuracy in our setting, obtaining up to 21 pt gain over the baselines.

## 1 Introduction

Neural networks have become the de-facto standard for solving perceptual tasks over low level representations, such as pixels in an image or audio signals. Recent research has also explored their application for solving symbolic reasoning tasks, requiring higher level inferences, such as neural theorem proving (Rocktäschel et al., 2015; Evans & Grefenstette, 2018; Minervini et al., 2020), and playing blocks world (Dong et al., 2019). The advantage of neural models for these tasks is that it will create a unified, end-to-end trainable representation for integrated AI systems that combine perceptual and high level reasoning. Our paper focuses on one such high level reasoning task – solving combinatorial problems in structured output spaces, e.g., solving a Sudoku or N-Queens puzzle. These can be thought of as Constraint Satisfaction problems (CSPs) where the underlying constraints are not explicitly available, and need to be learned from training data. We focus on learning such constraints by a non-autoregressive neural model where variables in the structured output space are decoded simultaneously (and therefore independently). Notably, most of the current state-of-the-art neural models for solving combinatorial problems, *e.g.*, SATNET (Wang et al., 2019), RRN (Palm et al., 2018), NLM (Dong et al., 2019), work with non autoregressive architectures because of their high efficiency of training and inference, since they do not have to decode the solution sequentially.

One of the key characteristics of such problems is *solution multiplicity* – there could be many correct solutions for any given input, even though we may be interested in finding any one of these solutions. For example, in a game of Sudoku with only 16 digits filled, there are always multiple correct solutions (McGuire et al., 2012), and obtaining any one of them suffices for solving Sudoku. Unfortunately, existing literature has completely ignored solution multiplicity, resulting in sub-optimally trained

---

[*]Equal contribution. Work done while at IIT Delhi. Current email: deepanshu.jindal@alumni.iitd.ac.in

networks. Our preliminary analysis of a state-of-the-art neural Sudoku solver (Palm et al., 2018)[1], which trains and tests on instances with single solutions, showed that it achieves a high accuracy of 96% on instances with single solution, but the accuracy drops to less than 25%, when tested on inputs that have multiple solutions. Intuitively, the challenge comes from the fact that (a) there could be a very large number of possible solutions for a given input, and (b) the solutions may be highly varied. For example, a 16-givens Sudoku puzzle could have as many as 10,000 solutions, with maximum hamming distance between any two solutions being 61. Hence, we argue that an explicit modeling effort is required to represent this solution multiplicity.

As the first contribution of our work, we formally define the novel problem of *One-of-Many Learning* (1oML). It is given training data of the form $\{(\mathbf{x_i}, \mathbf{Y_{x_i}})\}$, where $\mathbf{Y_{x_i}}$ denotes a subset of all correct outputs $\mathcal{Y}_{\mathbf{x_i}}$ associated with input $\mathbf{x_i}$. The goal of 1oML is to learn a function $f$ such that, for any input $\mathbf{x}$, $f(\mathbf{x}) = \mathbf{y}$ for some $\mathbf{y} \in \mathcal{Y}_{\mathbf{x}}$. We show that a naïve strategy that uses separate loss terms for each $(\mathbf{x_i}, \mathbf{y_{ij}})$ pair where $\mathbf{y_{ij}} \in \mathbf{Y_{x_i}}$ can result in a bad likelihood objective. Next, we introduce a multiplicity aware loss (CC-LOSS) and demonstrate its limitations for non-autoregressive models on structured output spaces. In response, we present our first-cut approach, MINLOSS, which picks up the single $\mathbf{y_{ij}}$ closest to the prediction $\hat{\mathbf{y}}_i$ based on the current parameters of prediction network (base architecture for function $f$), and uses it to compute and back-propagate the loss for that training sample $\mathbf{x_i}$. Though significantly better than naïve training, through a simple example, we demonstrate that MINLOSS can be sub-optimal in certain scenarios, due to its inability to pick a $\mathbf{y_{ij}}$ based on global characteristics of solution space.

To alleviate the issues with MINLOSS, we present two exploration based techniques, I-EXPLR and SELECTR, that select a $\mathbf{y_{ij}}$ in a non-greedy fashion, unlike MINLOSS. Both techniques are generic in the sense that they can work with any prediction network for the given problem. I-EXPLR relies on the prediction network itself for selecting $\mathbf{y_{ij}}$, whereas SELECTR is an RL based learning framework which uses a selection module to decide which $\mathbf{y_{ij}}$ should be picked for a given input $\mathbf{x_i}$, for back-propagating the loss in the next iteration. The SELECTR's selection module is trained jointly along with the prediction network using reinforcement learning, thus allowing us to trade-off exploration and exploitation in selecting the optimum $\mathbf{y_{ij}}$ by learning a probability distribution over the space of possible $\mathbf{y_{ij}}$'s for any given input $\mathbf{x_i}$.

We experiment on three CSPs: N-Queens, Futoshiki, and Sudoku. Our prediction networks for the first two problems are constructed using Neural Logic Machines (Dong et al., 2019), and for Sudoku, we use a state-of-the-art neural solver based on Recurrent Relational Networks (Palm et al., 2018). In all three problems, our experiments demonstrate that SELECTR vastly outperforms naïve baselines by up to 21 pts, underscoring the value of explicitly modeling solution multiplicity. SELECTR also consistently improves on other multiplicity aware methods, viz. CC-LOSS, MINLOSS, and I-EXPLR.

## 2 BACKGROUND AND RELATED WORK

**Related ML Models:** There are a few learning scenarios within weak supervision which may appear similar to the setting of 1oML, but are actually different from it. We first discuss them briefly. '*Partial Label Learning*' (PLL) (Jin & Ghahramani, 2002; Cour et al., 2011; Xu et al., 2019; Feng & An, 2019; Cabannes et al., 2020) involves learning from the training data where, for each input, a noisy set of candidate labels is given amongst which only one label is correct. This is different from 1oML in which there is no training noise and all the solutions in the solution set $\mathbf{Y_x}$ for a given $\mathbf{x}$ are correct. Though some of the recent approaches to tackle ambiguity in PLL (Cabannes et al., 2020) may be similar to our methods, i.e., MINLOSS , by the way of deciding which solution in the target set should be picked next for training, the motivations are quite different. Similarly, in the older work by (Jin & Ghahramani, 2002), the *EM model*, where the loss for each candidate is weighted by the probability assigned to that candidate by the model itself, can be seen as a naïve exploration based approach, applied to a very different setting. In PLL, the objective is to select the correct label out of many incorrect ones to reduce training noise, whereas in 1oML, selecting only one label for training provably improves the learnability and there is no question of reducing noise as all the labels are correct. Further, most of the previous work on PLL considers classification over a discrete output space with, say, $L$ labels, where as in 1oML, we work with structured output spaces, e.g., an $r$ dimensional vector space where each dimension represents a discrete space of $L$ labels. This

---

[1]Available at `https://data.dgl.ai/models/rrn-sudoku.pkl`

exponentially increases the size of the output space, making it intractable to enumerate all possible solutions as is typically done in existing approaches for PLL (Jin & Ghahramani, 2002).

Within weak supervision, the work on '*Multi Instance Learning*' (MIL) approach for Relation Extraction (RE) employs a selection module to pick a set of sentences to be used for training a relation classifier, given a set of noisy relation labels (Feng et al., 2018; Qin et al., 2018). This is different from us where multiplicity is associated with any given input, not with a class (relation).

Other than weak supervision, 1oML should also not be confused with the problems in the space of multi-label learning (Tsoumakas & Katakis, 2007). In multi-label learning, given a solution set $\mathbf{Y_x}$ for each input $\mathbf{x}$, the goal is to correctly predict each possible solution in the set $\mathbf{Y_x}$ for $\mathbf{x}$. Typically, a classifier is learned for each of the possible labels separately. On the other hand, in 1oML, the objective is to learn any one of the correct solutions for a given input, and a single classifier is learned. The characteristics of the two problems are quite different, and hence, also the solution approaches. As we show later, the two settings lead to requirements for different kinds of generalization losses.

**Solution Multiplicity in Other Settings:** There is some prior work related to our problem of solution multiplicity, albeit in different settings. An example is the task of video-prediction, where there can be multiple next frames ($\mathbf{y_{ij}}$) for a given partial video $\mathbf{x_i}$ (Henaff et al., 2017; Denton & Fergus, 2018). The multiplicity of solutions here arises from the underlying uncertainty rather than as a inherent characteristic of the domain itself. Current approaches model the final prediction as a combination of the deterministic part oblivious to uncertainty, and a non-determinstic part caused by uncertainty. There is no such separation in our case since each solution is inherently different from others.

Another line of work, which comes close to ours is the task of Neural Program Synthesis (Devlin et al., 2017; Bunel et al., 2018). Given a set of Input-Output (IO) pairs, the goal is to generate a valid program conforming to the IO specifications. For a given IO pair, there could be multiple valid programs, and often, training data may only have one (or a few) of them. Bunel et al. (2018) propose a solution where they define an alternate RL based loss using the correctness of the generated program on a subset of held out IO pairs as reward. In our setting, in the absence of the constraints (or rules) of the CSP, there is no such additional signal available for training outside the subset of targets $\mathbf{Y_x}$ for an input $\mathbf{x}$.

It would also be worthwhile to mention other tasks such as Neural Machine translation (Bahdanau et al., 2015; Sutskever et al., 2014), Summarization (Nallapati et al., 2017; Paulus et al., 2018), Image Captioning (Vinyals et al., 2017; You et al., 2016) *etc.*, where one would expect to have multiple valid solutions for any given input. *E.g.*, for a given sentence in language A, there could be multiple valid translations in language B. To the best of our knowledge, existing literature ignores solution multiplicity in such problems, and simply trains on all possible given labels for any given input.

**Models for Symbolic Reasoning:** Our work follows the line of recent research, which proposes neural architectures for implicit symbolic and relational reasoning problems (Santoro et al., 2018; Palm et al., 2018; Wang et al., 2019; Dong et al., 2019). We experiment with two architectures as base prediction networks: Neural Logic Machines (NLMs) (Dong et al., 2019), and Recurrent Relational Networks (RRNs) (Palm et al., 2018). NLMs allow learning of first-order logic rules expressed as Horn Clauses over a set of predicates, making them amenable to transfer over different domain sizes. The rules are instantiated over a given set of objects, where the groundings are represented as tensors in the neural space over which logical rules operate. RRNs use a graph neural network to learn relationships between symbols represented as nodes in the graph, and have been shown to be good at problems that require multiple steps of symbolic reasoning.

## 3 THEORY AND ALGORITHM

### 3.1 PROBLEM DEFINITION

**Notation:** Each possible solution (target) for an input (query) $\mathbf{x}$ is denoted by an r-dimensional vector $\mathbf{y} \in \mathcal{V}^r$, where each element of $\mathbf{y}$ takes values from a discrete space denoted by $\mathcal{V}$. Let $\mathcal{Y} = \mathcal{V}^r$, and let $\mathcal{Y}_\mathbf{x}$ denote the set of all solutions associated with input $\mathbf{x}$. We will use the term *solution multiplicity* to refer to the fact that there could be multiple possible solutions $\mathbf{y}$ for a given input $\mathbf{x}$. In our setting, the solutions in $\mathcal{Y}_\mathbf{x}$ span a structured combinatorial subspace of $\mathcal{V}^r$, and can be thought of as representing solutions to an underlying Constraint Satisfaction Problem (CSP). For example in N-Queens, $\mathbf{x}$ would denote a partially filled board, and $\mathbf{y}$ denote a solution for the input board.

Given a set of inputs $\mathbf{x_i}$ along with a subset of associated solutions $\mathbf{Y_{x_i}} \subseteq \mathcal{Y}_{\mathbf{x_i}}$, *i.e.*, given a set of $(\mathbf{x_i}, \mathbf{Y_{x_i}})$ pairs, we are interested in learning a mapping from $\mathbf{x}$ to any one $\mathbf{y}$ among many possible solutions for $\mathbf{x}$. Formally, we define the *One-of-Many-Learning (1oML)* problem as follows.

**Definition 1.** *Given training data $\mathbb{D}$ of the form, $\{(\mathbf{x_i}, \mathbf{Y_{x_i}})\}_{i=1}^{m}$, where $\mathbf{Y_{x_i}}$ denotes a subset of solutions associated with input $\mathbf{x_i}$, and $m$ is the size of training dataset,* One-of-Many-Learning (1oML) *is defined as the problem of learning a function $f$ such that, for any input $\mathbf{x}$, $f(\mathbf{x}) = \mathbf{y}$ for some $\mathbf{y} \in \mathcal{Y}_{\mathbf{x}}$, where $\mathcal{Y}_{\mathbf{x}}$ is the set of all solutions associated with $\mathbf{x}$.*

We use parameterized neural networks to represent our mapping function. We use $M_\Theta$ to denote a non-autoregressive network $M$ with associated set of parameters $\Theta$. We use $\hat{\mathbf{y}}_{\mathbf{i}}$ ($\hat{\mathbf{y}}$) to denote the network output corresponding to input $\mathbf{x_i}$ ($\mathbf{x}$), *i.e.*, $\hat{\mathbf{y}}_{\mathbf{i}}$ ($\hat{\mathbf{y}}$) is the $\arg\max$ of the learnt conditional distribution over the output space $\mathcal{Y}$ given the input $\mathbf{x_i}$ ($\mathbf{x}$). We are interested in finding a $\Theta^*$ that solves the 1oML problem as defined above. Next, we consider various formulations for the same.

### 3.2 OBJECTIVE FUNCTION

**Naïve Objective**: In the absence of solution multiplicity, *i.e.* when target set $\mathbf{Y_{x_i}} = \{\mathbf{y_i}\}, \forall\mathbf{i}$, the standard method to train such models is to minimize the total loss, $L(\Theta) = \sum_{i=1}^{m} l_\Theta(\hat{\mathbf{y}}_{\mathbf{i}}, \mathbf{y_i})$, where $l_\Theta(\hat{\mathbf{y}}_{\mathbf{i}}, \mathbf{y_i})$ is the loss between the prediction $\hat{\mathbf{y}}_{\mathbf{i}}$ and the unique target $\mathbf{y_i}$ for the input $\mathbf{x_i}$. We find the optimal $\Theta^*$ as $\arg\min_\Theta L(\Theta)$. A Naïve extension of this for 1oML would be to sum the loss over all targets in $\mathbf{Y_x}$, *i.e.*, minimize the following loss function:

$$L(\Theta) = \frac{1}{m} \sum_{i=1}^{m} \sum_{\mathbf{y_{ij}} \in \mathbf{Y_{x_i}}} l_\Theta(\hat{\mathbf{y}}_{\mathbf{i}}, \mathbf{y_{ij}}) \tag{1}$$

We observe that loss function in eq. (1) would unnecessarily penalize the model when dealing with solution multiplicity. Even when it is correctly predicting one of the targets for an input $\mathbf{x_i}$, the loss with respect to the other targets in $\mathbf{Y_{x_i}}$ could be rather high, hence misguiding the training process. Example 1 below demonstrates such a case. For illustration, we will use the cross-entropy loss, *i.e.*, $l_\Theta(\hat{\mathbf{y}}, \mathbf{y}) = -\sum_k \sum_l \mathbb{1}\{\mathbf{y}[k] = v_l\} \log(P(\hat{\mathbf{y}}[k] = v_l))$, where $v_l \in \mathcal{V}$ varies over the elements of $\mathcal{V}$, and $k$ indices over $r$ dimensions in the solution space. $\mathbf{y}[k]$ denotes the $k^{th}$ element of $\mathbf{y}$.

**Example 1.** *Consider a learning problem over a discrete (Boolean) input space $\mathcal{X} = \{0, 1\}$ and Boolean target space in two dimensions, i.e., $\mathcal{Y} = \mathcal{V}^r = \{0, 1\}^2$. Let this be a trivial learning problem where $\forall\mathbf{x}$, the solution set is $\mathbf{Y_x} = \{(0, 1), (1, 0)\}$. Then, given a set of examples $\{\mathbf{x_i}, \mathbf{Y_{x_i}}\}$, the Naïve objective (with $l_\Theta$ as cross entropy) will be minimized, when $P(\hat{\mathbf{y}}_{\mathbf{i}}[k] = 0) = P(\hat{\mathbf{y}}_{\mathbf{i}}[k] = 1) = 0.5$, for $k \in \{1, 2\}, \forall\mathbf{i}$, which can not recover either of the desired solutions: $(0, 1)$ or $(1, 0)$.*

The problem arises from the fact that when dealing with 1oML, the training loss defined in eq. (1) is no longer a consistent predictor of the generalization error as formalized below.

**Lemma 1.** *The training loss $L(\Theta)$ as defined in eq. (1) is an inconsistent estimator of generalization error for 1oML, when $l_\Theta$ is a zero-one loss, i.e., $l_\Theta(\hat{\mathbf{y}}_{\mathbf{i}}, \mathbf{y_{ij}}) = \mathbb{1}\{\hat{\mathbf{y}}_{\mathbf{i}} \neq \mathbf{y_{ij}}\}$. (Proof in Appendix).*

For the task of PLL, Jin & Ghahramani (2002) propose a modification of the cross entropy loss to tackle multiplicity of labels in the training data. Instead of adding the log probabilities, it maximizes the log of total probability over the given target set. Inspired by Feng et al. (2020), we call it CC-Loss: $L_{cc}(\Theta) = -\frac{1}{m} \sum_{i=1}^{m} \log\left(\sum_{\mathbf{y_{ij}} \in \mathbf{Y_{x_i}}} Pr(\mathbf{y_{ij}}|\mathbf{x_i}; \Theta)\right)$. However, in the case of structured prediction, optimizing $L_{cc}$ requires careful implementation due to its numerical instability (see Appendix). Moreover, for non-autoregressive models, CC-Loss also suffers from the same issues illustrated in example 1 for naïve objective.

**New Objective:** We now motivate a better objective function based on an unbiased estimator. In general, we would like $M_\Theta$ to learn a conditional probability distribution $Pr(\mathbf{y}|\mathbf{x_i}; \Theta)$ over the output space $\mathcal{Y}$ such that the entire probability mass is concentrated on the desired solution set $\mathbf{Y_{x_i}}$, *i.e.*, $\sum_{\mathbf{y_{ij}} \in \mathbf{Y_{x_i}}} Pr(\mathbf{y_{ij}}|\mathbf{x_i}; \Theta) = 1, \forall\mathbf{i}$. If such a conditional distribution is learnt, then we can easily sample a $\mathbf{y_{ij}} \in \mathbf{Y_{x_i}}$ from it. CC-Loss is indeed trying to achieve this. However, ours being a structured output space, it is intractable to represent all possible joint distributions over the possible solutions in $\mathbf{Y_{x_i}}$, especially for non-autoregressive models[2].

---

[2]Autoregressive models may have the capacity to represent certain class of non-trivial joint distributions, *e.g.*, $Pr(y[1], y[2]|x)$ could be modeled as $Pr(y[1]|x)Pr(y[2]|y[1]; x)$, but requires sequential decoding during inference. Studying the impact of solution multiplicity on autoregressive models is beyond the current scope.

Hence, we instead design a loss function which forces the model to learn a distribution in which the probability mass is concentrated on *any one* of the targets $\mathbf{y_{ij}} \in \mathbf{Y_{x_i}}$. We call such distributions as one-hot. To do this, we introduce $|\mathbf{Y_{x_i}}|$ number of new learnable Boolean parameters, $\mathbf{w_i}$, for each query $\mathbf{x_i}$ in the training data, and correspondingly define the following loss function:

$$L_\mathbf{w}(\Theta, \mathbf{w}) = \frac{1}{m} \sum_{i=1}^{m} \sum_{\mathbf{y_{ij}} \in \mathbf{Y_{x_i}}} w_{\mathbf{ij}} l_\Theta(\hat{\mathbf{y}}_\mathbf{i}, \mathbf{y_{ij}}) \tag{2}$$

Here, $w_{\mathbf{ij}} \in \{0, 1\}$ and $\sum_\mathbf{j} w_{\mathbf{ij}} = 1, \forall \mathbf{i}$, where $\mathbf{j}$ indices over solutions $\mathbf{y_{ij}} \in \mathbf{Y_{x_i}}$. The last constraint over Boolean variables $w_{\mathbf{ij}}$ enforces that exactly one of the weights in $\mathbf{w_i}$ is 1 and all others are zero.

**Lemma 2.** *Under the assumption $\mathbf{Y_{x_i}} = \mathcal{Y}_{\mathbf{x_i}}, \forall \mathbf{i}$, the loss $L'(\Theta) = \min_\mathbf{w} L_\mathbf{w}(\Theta, \mathbf{w})$, defined as the minimum value of $L_\mathbf{w}(\Theta, \mathbf{w})$ (defined in eq. (2)) with respect to $\mathbf{w}$, is a consistent estimator of generalization error for 1oML, when $l_\Theta$ is a zero-one loss, i.e., $l_\Theta(\hat{\mathbf{y}}_\mathbf{i}, \mathbf{y_{ij}}) = \mathbb{1}\{\hat{\mathbf{y}}_\mathbf{i} \neq \mathbf{y_{ij}}\}$.*

We refer to Appendix for details. Next, we define our new objective as:

$$\min_{\Theta, \mathbf{w}} L_\mathbf{w}(\Theta, \mathbf{w}) \text{ s.t. } w_{\mathbf{ij}} \in \{0, 1\} \, \forall \mathbf{i}, \forall \mathbf{j} \text{ and } \sum_{\mathbf{j}=1}^{|\mathbf{Y_{x_i}}|} w_{\mathbf{ij}} = 1, \forall \mathbf{i} = 1 \dots m \tag{3}$$

### 3.3 Greedy Formulation: MinLoss

In this section, we present one possible way to optimize our desired objective $\min_{\Theta, \mathbf{w}} L_\mathbf{w}(\Theta, \mathbf{w})$. It alternates between optimizing over the $\Theta$ parameters, and optimizing over $\mathbf{w}$ parameters. While $\Theta$ parameters are optimized using SGD, the weights $\mathbf{w}$ are selected greedily for a given $\Theta = \Theta^t$ at each iteration, *i.e.*, it assigns a non-zero weight to the solution corresponding to the minimum loss amongst all the possible $\mathbf{y_{ij}} \in \mathbf{Y_{x_i}}$ for each $\mathbf{i} = 1 \dots m$:

$$w_{\mathbf{ij}}^{(t)} = \mathbb{1}\left\{\mathbf{y_{ij}} = \operatorname*{argmin}_{\mathbf{y} \in \mathbf{Y_{x_i}}} l_{\Theta^{(t)}}\left(\hat{\mathbf{y}}_\mathbf{i}^{(t)}, \mathbf{y}\right)\right\}, \forall \mathbf{i} = 1 \dots m \tag{4}$$

This can be done by computing the loss with respect to each target, and picking the one which has the minimum loss. We refer to this approach as MinLoss. Intuitively, for a given set of $\Theta^{(t)}$ parameters, MinLoss greedily picks the weight vector $\mathbf{w_i}^{(t)}$, and uses them to get the next set of $\Theta^{(t+1)}$ parameters using SGD update.

$$\Theta^{(t+1)} \leftarrow \Theta^{(t)} - \alpha_\Theta \nabla_\Theta L_\mathbf{w}(\Theta, \mathbf{w})|_{\Theta=\Theta^{(t)}, \mathbf{w}=\mathbf{w}^{(t)}} \tag{5}$$

One significant challenge with MinLoss is the fact that it chooses the current set of $\mathbf{w}$ parameters independently for each example based on current $\Theta$ values. While this way of picking the $\mathbf{w}$ parameters is optimal if $\Theta$ has reached the optima, *i.e.* $\Theta = \Theta^*$, it can lead to sub-optimal choices when both $\Theta$ and $\mathbf{w}$ are being simultaneously trained. Following example illustrates this.

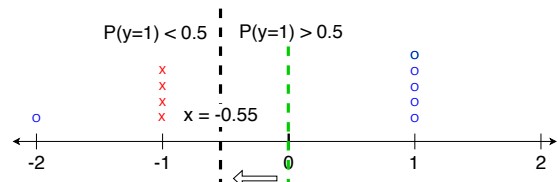

Figure 1: Decision Boundary learnt by logistic regression guided by MinLoss. Green line at $\mathbf{x} = 0$ is the initial decision boundary and black vertical line at $\mathbf{x} = -0.55$ is the decision boundary at convergence.

**Example 2.** *Consider a simple task with a one-dimensional continuous input space $\mathcal{X} \subset \mathcal{R}$, and target space $\mathcal{Y} = \{0, 1\}$. Consider learning with 10 examples, given as $(\mathbf{x} = 1, \mathbf{Y_x} = \{1\})$ (5 examples), $(\mathbf{x} = -1, \mathbf{Y_x} = \{0, 1\})$ (4 examples), $(\mathbf{x} = -2, \mathbf{Y_x} = \{1\})$ (1 example). The optimal decision hypothesis is given as: $\mathbf{y} = \mathbb{1}\{\mathbf{x} > \alpha\}$, for $\alpha \leq -2$, or $\mathbf{y} = \mathbb{1}\{\mathbf{x} < \beta\}$, for $\beta \geq 1$. Assume learning this with logistic regression using MinLoss as the training algorithm optimizing the objective in eq. (3). If we initialize the parameters of logistic such that the starting hypothesis is given by $\mathbf{y} = \mathbb{1}\{\mathbf{x} > 0\}$ (logistic parameters: $\theta_1 = 0.1$, $\theta_0 = 0$), MinLoss will greedily pick the target $\mathbf{y} = 0$ for samples with $\mathbf{x} = -1$, repeatedly. This will result in the learning algorithm converging to the decision hypothesis $\mathbf{y} = \mathbb{1}\{\mathbf{x} > -0.55\}$, which is sub-optimal since the input with $\mathbf{x} = -2$ is incorrectly classified (fig. 1, see Appendix for a detailed discussion).*

MinLoss is not able to achieve the optimum since it greedily picks the target for each query $\mathbf{x_i}$ based on current set of parameters and gets stuck in local mimima. This is addressed in the next section.

### 3.4 REINFORCEMENT LEARNING FORMULATION: SELECTR

In this section, we will design a training algorithm that fixes some of the issues observed with MINLOSS. Considering the Example 2 above, the main problem with MINLOSS is its inability to consider alternate targets which may not be greedily optimal at the current set of parameters. A better strategy will try to explore alternative solutions as a way of reaching better optima, e.g., in example 2 we could pick, for the input $\mathbf{x} = -1$, the target $\mathbf{y} = 1$ with some non-zero probability, to come out of the local optima. In the above case, this also happens to be the globally optimal strategy. This is the key motivation for our RL-based strategy proposed below.

A natural questions arises: how should we assign the probability of picking a particular target? A naïve approach would use the probability assigned by the underlying $M_\Theta$ network as a way of deciding the amount of exploration on each target $\mathbf{y}$. We call it I-EXPLR. We argue below why this may not always be an optimal choice.

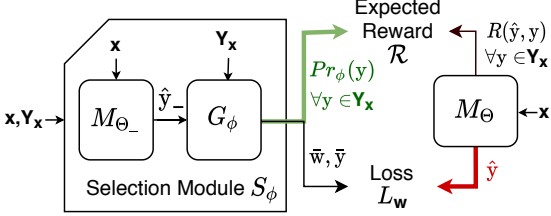

Figure 2: Flow-diagram for our RL Framework

We note that the amount of exploration required may depend in complex ways on the global solution landscape, as well as the current set of parameters. Therefore, we propose a strategy, which makes use of a separate *selection module* (a neural network), which takes as input, the current example $(\mathbf{x_i}, \mathbf{Y_{x_i}})$, and outputs the probability of picking each target for training $\Theta$ in the next iteration. Our strategy is RL-based since, we can think of choosing each target (for a given input) as an action that our selection module needs to take. Our selection module is trained using a reward that captures the quality of selecting the corresponding target for training the prediction network. We next describe its details.

**Selection Module ($S_\phi$):** This is an RL agent or a policy network where the action is to select a target, $\mathbf{y_{ij}} \in \mathbf{Y_{x_i}}$, for each $\mathbf{x_i}$. Given a training sample, $(\mathbf{x_i}, \mathbf{Y_{x_i}})$, it first internally predicts $\hat{\mathbf{y}}_{\mathbf{i}_-} = M_{\Theta_-}(\mathbf{x_i})$, using a past copy of the parameters $\Theta_-$. This prediction is then fed as an input along with the target set, $\mathbf{Y_{x_i}}$, to a latent model, $G_\phi$, which outputs a probability distribution $Pr_\phi(\mathbf{y_{ij}}), \forall \mathbf{y_{ij}} \in \mathbf{Y_{x_i}}$, s.t. $\sum_{\mathbf{y_{ij}}} Pr_\phi(\mathbf{y_{ij}}) = 1$. $S_\phi$ then picks a target $\bar{\mathbf{y}}_\mathbf{i} \in \mathbf{Y_{x_i}}$ based on the distribution $Pr_\phi(\mathbf{y_{ij}})$ and returns a $\bar{\mathbf{w}}_\mathbf{i}$ such that $\forall \mathbf{i}, \bar{\mathbf{w}}_{\mathbf{ij}} = 1$ if $\mathbf{y_{ij}} = \bar{\mathbf{y}}_\mathbf{i}$, and $\bar{\mathbf{w}}_{\mathbf{ij}} = 0$ otherwise.

**Update of $\phi$ Parameters:** The job of the selection module is to pick one target, $\bar{\mathbf{y}}_\mathbf{i} \in \mathbf{Y_{x_i}}$, for each input $\mathbf{x_i}$, for training the prediction network $M_\Theta$. If we were given an oracle to tell us which $\bar{\mathbf{y}}_\mathbf{i}$ is most suited for training $M_\Theta$, we would have trained the selection module $S_\phi$ to match the oracle. In the absence of such an oracle, we train $S_\phi$ using a reward scheme. Intuitively, $\bar{\mathbf{y}}_\mathbf{i}$ would be a good choice for training $M_\Theta$, if it is "easier" for the model to learn to predict $\bar{\mathbf{y}}_\mathbf{i}$. In our reward design, we measure this degree of ease using hamming distance between $\bar{\mathbf{y}}_\mathbf{i}$ and $M_\Theta$'s prediction $\hat{\mathbf{y}}_\mathbf{i}$, *i.e.*, $R(\hat{\mathbf{y}}_\mathbf{i}, \bar{\mathbf{y}}_\mathbf{i}) = \sum_{k=1}^{r} \mathbb{1}\{\hat{\mathbf{y}}_\mathbf{i}[k] = \bar{\mathbf{y}}_\mathbf{i}[k]\}$. We note that there are other choices as well for the reward, *e.g.*, a binary reward, which gives a positive reward of 1 only if the prediction model $M_\Theta$ has learnt to predict the selected target $\bar{\mathbf{y}}_\mathbf{i}$. Our reward scheme is a granular proxy of this binary reward and makes it easier to get a partial reward even when the binary reward would be 0.

The expected reward for RL can then be written as:
$$\mathcal{R}(\phi) = \sum_{\mathbf{i}=1}^{m} \sum_{\mathbf{y_{ij}} \in \mathbf{Y_{x_i}}} Pr_\phi(\mathbf{y_{ij}}) R(\hat{\mathbf{y}}_\mathbf{i}, \mathbf{y_{ij}}) \tag{6}$$

We make use of policy gradient to compute the derivative of the expected reward with respect to the $\phi$ parameters. Accordingly, update equation for $\phi$ can be written as:
$$\phi^{(t+1)} \leftarrow \phi^{(t)} + \alpha_\phi \nabla_\phi \mathcal{R}(\phi)|_{\phi=\phi^{(t)}} \tag{7}$$

**Update of $\Theta$ Parameters:** Next step is to use the output of the selection module, $\bar{\mathbf{w}}_\mathbf{i}$ corresponding to the sampled target $\bar{\mathbf{y}}_\mathbf{i}, \forall \mathbf{i}$, to train the $M_\Theta$ network. The update equation for updating the $\Theta$ parameters during next learning iteration can be written as:
$$\Theta^{(t+1)} \leftarrow \Theta^{(t)} - \alpha_\Theta \nabla_\Theta L_{\mathbf{w}}(\Theta, \mathbf{w})|_{\Theta=\Theta^{(t)}, \mathbf{w}=\bar{\mathbf{w}}^{(t)}} \tag{8}$$

Instead of backpropagating the loss gradient at a sampled target $\bar{\mathbf{y}}_\mathbf{i}$, one could also backpropagate the gradient of the expected loss given the distribution $Pr_\phi(\mathbf{y_{ij}})$. In our experiments, we backpropagate

through the expected loss since our action space for the selection module $S_\phi$ is tractable. Figure 2 represents the overall framework. In the diagram, gradients for updating $\Theta$ flow back through the red line and gradients for updating $\phi$ flow back through the green line.

## 3.5 TRAINING ALGORITHM

We put all the update equations together and describe the key components of our training algorithm below. Algorithm 1 presents a detailed pseudocode.

---

**Algorithm 1** Joint Training of Prediction Network $M_\Theta$ & Selection Module $S_\phi$

---
1   $\Theta_0 \leftarrow$ **Pre-train** $\Theta$ using eq. (4) and eq. (5)
2   **In Selection Module (SM):** $\Theta_- \leftarrow \Theta_0$
3   $\phi_0 \leftarrow$ **Pre-train** $\phi$ using rewards from $M_\Theta$ in eq. (7)
4   Initialize: $t \leftarrow 0$
5   **while** *not converged* **do**
6      $B \leftarrow$ Randomly fetch a mini-batch
7      **for** $i \in B$ **do**
8         **Get weights:** $\mathbf{w_i} \leftarrow S_\phi((\mathbf{x_i}, \mathbf{Y_{x_i}}), \Theta_-)$
9         **Get model predictions:** $\hat{\mathbf{y}}_\mathbf{i} \leftarrow M_{\Theta^t}(\mathbf{x_i})$
10        **Get rewards:** $\mathbf{r_i} \leftarrow [R(\hat{\mathbf{y}}_\mathbf{i}, \mathbf{y_{ij}}), \forall \mathbf{y_{ij}} \in \mathbf{Y_{x_i}}]$
     **end**
11      **Update** $\phi$: Use eq. (7) to get $\phi^{(t+1)}$
12      **Update** $\Theta$: Use eq. (8) to get $\Theta^{(t+1)}$
13      **Update** $\Theta_- \leftarrow \Theta^{(t+1)}$ if $t\%copyitr = 0$ (in SM)
14      **Increment** $t \leftarrow t + 1$
  **end**

---

**Pre-training:** It is a common strategy in many RL based approaches to first pre-train the network weights using a simple strategy. Accordingly, we pre-train both the $M_\Theta$ and $S_\phi$ networks before going into joint training. First, we pre-train $M_\Theta$. In our experiments, we observe that in some cases, pre-training $M_\Theta$ using only those samples from training data $\mathbb{D}$ for which there is only a unique solution, *i.e.*, $\{(\mathbf{x_i}, \mathbf{Y_{x_i}}) \in \mathbb{D}$ s.t. $|\mathbf{Y_{x_i}}| = 1\}$ gives better performance than pre-training with MIN-LOSS. Therefore, we pre-train using both the approaches and select the better one based on their performance on a held out dev set. Once the prediction network is pre-trained, a copy of it is given to the selection module to initialize $M_{\Theta_-}$. Keeping $\Theta$ and $\Theta_-$ fixed and identical to each other, the latent model, $G_\phi$, in the selection module is pre-trained using the rewards given by the pre-trained $M_\Theta$ and the internal predictions given by $M_{\Theta_-}$.

**Joint Training:** After pre-training, both prediction network $M_\Theta$ and selection module $S_\phi$ are trained jointly. In each iteration $t$, selection module first computes the weights, $\bar{\mathbf{w}}_\mathbf{i}^t$, for each sample in the mini-batch. The prediction network computes the prediction $\hat{\mathbf{y}}_\mathbf{i}^t$ and rewards $R(\hat{\mathbf{y}}_\mathbf{i}^t, \mathbf{y_{ij}}), \forall \mathbf{y_{ij}} \in \mathbf{Y_{x_i}}$. The parameters $\phi^t$ and $\Theta^t$ are updated simultaneously using eq. (7) and eq. (8), respectively. The copy of the prediction network within selection module, *i.e.*, $M_{\Theta_-}$ in $S_\phi$, is updated with the latest parameters $\Theta^t$ after every $copyitr$ updates where $copyitr$ is a hyper-parameter.

## 4 EXPERIMENTS

The main goal of our experiments is to evaluate the four multiplicity aware methods: CC-LOSS, MINLOSS, informed exploration (I-EXPLR) and RL based exploration (SELECTR), when compared to baseline approaches that completely disregard the problem of solution multiplicity. We also wish to assess the performance gap, if any, between queries with a unique solution and those with many possible solutions. To answer these questions, we conduct experiments on three different tasks (N-Queens, Futoshiki & Sudoku), trained over two different prediction networks, as described below.[3]

### 4.1 DATASETS AND PREDICTION NETWORKS

**N-Queens:** Given a query, *i.e.*, a chess-board of size $N \times N$ and a placement of $k < N$ non-attacking queens on it, the task of N Queens is to place the remaining $N - k$ queens, such that no two queens are attacking each other. We train a Neural Logic Machine (NLM) model (Dong et al., 2019) as the prediction network $M_\Theta$ for solving queries for this task. To model N-Queens within NLM, we represent a query $\mathbf{x}$ and the target $\mathbf{y}$ as $N^2$ dimensional Boolean vectors with 1 at locations where a Queen is placed. We use another smaller NLM architecture as the latent model $G_\phi$.

We train our model on 10–Queens puzzles and test on 11–Queens puzzles, both with 5 placed queens. This size-invariance in training and test is a key strength of NLM architecture, which we exploit in our experiments. To generate the train data, we start with all possible valid 10–Queens board configurations and randomly mask any 5 queens, and then check for all possible valid completions to

---

[3]Further details of software environments, hyperparameters and dataset generation are in the appendix.

generate potentially multiple solutions for an input. Test data is also generated similarly. Training and testing on different board sizes ensures that no direct information leaks from test to train. Queries with multiple solutions have 2-6 solutions, so we choose $\mathbf{Y_{x_i}} = \mathcal{Y}_{\mathbf{x_i}}, \forall \mathbf{x_i}$.

**Futoshiki:** This is a logic puzzle in which we are given a grid of size $N \times N$, and the goal is to fill the grid with digits from $\{1 \dots N\}$ such that no digit is repeated in a row or a column. $k$ out of $N^2$ positions are already filled in the input query $\mathbf{x}$ and the remaining $N^2 - k$ positions need to be filled. Further, inequality constraints are specified between some pairs of adjacent grid positions, which need to be honored in the solution. Our prediction network, and latent model use NLM, and the details (described in Appendix) are very similar to that of N–Queens.

Similar to N–Queens, we do size-invariant training – we train our models on $5 \times 5$ puzzles with $14$ missing digits and test on $6 \times 6$ puzzles with $20$ missing digits. Similar to N–Queens, we generate all possible valid grids and randomly mask out the requisite number of digits to generate train and test data. For both train and test queries we keep up to five inequality constraints of each type: $>$ and $<$.

**Sudoku:** We also experiment on Sudoku, which has been used as the task of choice for many recent neural reasoning works (Palm et al., 2018; Wang et al., 2019). We use Relational Recurrent Networks (RRN) (Palm et al., 2018) as the prediction network since it has recently shown state-of-the-art performance on the task. We use a 5 layer CNN as our latent model $G_\phi$. Existing Sudoku datasets (Royle, 2014; Park, 2018), do not expose the issues with solution multiplicity. In response, we generate our own dataset by starting with a collection of Sudoku puzzles with unique solutions that have 17 digits filled. We remove one of the digits, thus generating a puzzle, which is guaranteed to have solution multiplicity. We then randomly add 1 to 18 of the digits back from the solution of the original puzzle, while ensuring that the query continues to have more than 1 solution. This generates our set of multi-solution queries with a uniform distribution of filled digits from 17 to 34. We mix an equal number of unique solution queries (with same filled distribution). Because some $\mathbf{x_i}$s may have hundreds of solutions, we randomly sample 5 of them from $\mathcal{Y}_{\mathbf{x_i}}$, i.e., $|\mathbf{Y_{x_i}}| \leq 5$ in the train set. For each dataset, we generate a devset in a manner similar to the test set.

Table 1: Statistics of datasets. 'Train', 'Test' and task names are abbreviated. Devset similar to test.

|  | N-Qn (Tr) | N-Qn (Tst) | Futo. (Tr) | Futo. (Tst) | Sud. (Tr) | Sud. (Tst) |
|---|---|---|---|---|---|---|
| # of queries | 165,744 | 10,000 | 10,000 | 10,000 | 20,000 | 10,000 |
| %age of *MS* queries | 7.04% | 16.67% | 17.05% | 24.95% | 50% | 50% |
| Avg solns per *MS* query | 2.1 | 2.2 | 2.2 | 2.4 | 13.8 | 13.7 |

## 4.2 BASELINES AND EVALUATION METRIC

Our comparison baselines include: (1) *Naïve*: backpropagating $L(\Theta)$ through each solution independently using Equation (1), (2) *Unique*: computing $L(\Theta)$ only over the subset of training examples that have a unique solution, and (3) *Random*: backpropagating $L(\Theta)$ through one arbitrarily picked solution $\mathbf{y_i} \in \mathbf{Y_{x_i}}$ for every $\mathbf{x_i}$ in the train data, and keeping this choice fixed throughout the training.

We separately report performance on two mutually exclusive subsets of test data: *OS*: queries with a unique solution, and *MS*: those with multiple solutions. For all methods, we tune various hyperparameters (and do early stopping) based on the devset performance. Additional parameters for the four multiplicity aware methods include the ratio of *OS* and *MS* examples in training.[4] I-EXPLR and SELECTR also select the pre-training strategy as described in Section 3.5. For all tasks, we consider the output of a prediction network as correct only if it is a valid solution for the underlying CSP. No partial credit is given for guessing parts of the output correctly.

## 4.3 RESULTS AND DISCUSSION

We report the accuracies across all tasks and models in Table 2. For each setting, we report the mean over three random runs (with different seeds), and also the accuracy on the best of these runs selected via the devset (in the parentheses). We first observe that *Naïve* and *Random* perform significantly worse than *Unique* in all the tasks, not only on *MS*, but on *OS* as well. This suggests that, 1oML models that explicitly handle solution multiplicity, even if by simply discarding multiple solutions, are much better than those that do not recognize it at all.

---

[4]Futoshiki and N–Queens training datasets have significant *OS-MS* imbalance (see Table 1), necessitating managing this ratio by undersampling *OS*. This is similar to standard approach in class imbalance problems.

Table 2: Mean (Max) test accuracy over three runs for multiplicity aware methods compared with baselines. *OS*: test queries with only one solution, *MS*: queries with more than one solution.

| | | *Naïve* | *Random* | *Unique* | CC-Loss | MinLoss | I-Explr | SelectR |
|---|---|---|---|---|---|---|---|---|
| **N-Queens** | *OS* | 70.59 (70.56) | 72.91 (73.86) | 75.09 (75.76) | 75.31 (76.19) | 77.29 (78.00) | 77.35 (79.01) | **79.73 (80.12)** |
| | *MS* | 55.34 (60.97) | 61.13 (61.81) | 66.85 (69.48) | 75.76 (75.36) | 77.22 (77.82) | 79.46 (81.95) | **79.68 (82.37)** |
| | Overall | 68.04 (68.96) | 70.94 (71.85) | 73.72 (74.71) | 75.39 (76.05) | 77.28 (77.97) | 77.7 (79.50) | **79.72 (80.50)** |
| **Futoshiki** | *OS* | 65.59 (66.8) | 65.49 (65.22) | 67.63 (69.49) | 77.68 (78.36) | 76.78 (78.24) | **78.15 (77.96)** | 78.01 (78.36) |
| | *MS* | 14.99 (18.04) | 14.22 (18.84) | 19.13 (23.33) | 69.3 (68.62) | 70.35 (69.06) | 70.88 (73.71) | **71.57 (72.42)** |
| | Overall | 52.96 (54.63) | 52.7 (53.65) | 55.53 (57.97) | 75.59 (75.93) | 75.18 (75.95) | 76.33 (76.90) | **76.4 (76.88)** |
| **Sudoku** | *OS* | 87.85 (89.08) | 87.53 (86.24) | **89.19 (90.24)** | 88.26 (86.78) | 88.25 (88.22) | 88.73 (89.62) | 88.69 (87.94) |
| | *MS* | 09.13 (10.59) | 13.65 (16.07) | 66.39 (70.20) | 76.58 (78.38) | 76.93 (78.94) | 80.19 (81.45) | **81.73 (85.45)** |
| | Overall | 48.49 (49.84) | 50.59 (51.15) | 77.79 (80.22) | 82.42 (82.58) | 82.59 (83.58) | 84.46 (85.54) | **85.21 (86.70)** |

Predictably, all multiplicity aware methods vastly improve upon the performance of naïve baselines, with a dramatic 13-52 pt gains between *Unique* and SelectR on queries with multiple solutions.

Comparing MinLoss and SelectR, we find that our RL-based approach outperforms Min-Loss consistently, with p-values (computed using McNemar's test for the best models selected based on validation set) of $1.00\mathrm{e}{-16}$, $0.03$, and $1.69\mathrm{e}{-18}$ for NQueens, Futoshiki and Sudoku respectively (see Appendix for seed-wise comparisons of gains across tasks). On the other hand, informed exploration technique, I-Explr, though improves over MinLoss on two out of three tasks, it performs worse than SelectR in all the domains. This highlights the value of RL based exploration on top of the greedy target selection of MinLoss as well as over the simple exploration of I-Explr. We note that this is due to more exploratory power of SelectR over I-Explr. See Appendix for more discussion and experiments comparing the two exploration techniques.

Figure 3: Accuracy vs size of query's solution set (with $95\%$ confidence interval)

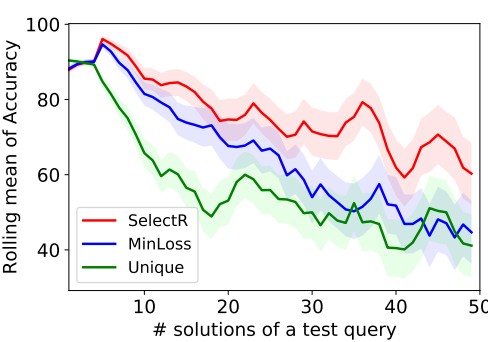

Recall that Sudoku training set has no more than 5 solutions for a query, irrespective of the actual number of solutions – i.e, for many $\mathbf{x_i}$, $\mathbf{Y_{x_i}} \subsetneq \mathcal{Y}_{\mathbf{x_i}}$. Despite incomplete solution set, significant improvement over baselines is obtained, indicating that our formulation handles solution multiplicity even with incomplete information. Furthermore, the large variation in the size of solution set ($|\mathcal{Y}_{\mathbf{x}}|$) in Sudoku allows us to assess its effect on the overall performance. We find that all models get worse as $|\mathcal{Y}_{\mathbf{x}}|$ increases (fig. 3), even though SelectR remains the most robust (see Appendix for details).

## 5 Conclusion and Future Work

In this paper, we have defined 1oML: the task of learning one of many solutions for combinatorial problems in structured output spaces. We have identified solution multiplicity as an important aspect of the problem, which if not handled properly, may result in sub-optimal models. As a first cut solution, we proposed a greedy approach: MinLoss formulation. We identified certain shortcomings with the greedy approach and proposed two exploration based formulations: I-Explr and an RL formulation, SelectR, which overcomes some of the issues in MinLoss by exploring the locally sub-optimal choices for better global optimization.

Experiments on three different tasks using two different prediction networks demonstrate the effectiveness of our approach in training robust models under solution multiplicity [5].

It is interesting to note that for traditional CSP solvers, *e.g.*(Selman et al., 1993; Mahajan et al., 2004), a problem with many solutions will be considered an easy problem, whereas for neural models, such problems appear much harder (Figure 3). As a future work, it will be interesting to combine symbolic CSP solvers with SelectR to design a much stronger neuro-symbolic reasoning model.

---

[5] All the code and datasets are available at: *https://sites.google.com/view/yatinnandwani/1oml*

## ACKNOWLEDGEMENT

We thank IIT Delhi HPC facility[6] for computational resources. We thank anonymous reviewers for their insightful comments and suggestions, in particular *AnonReviewer4* for suggesting a simple yet effective informed exploration strategy (I-EXPLR). Mausam is supported by grants from Google, Bloomberg, 1MG and Jai Gupta chair fellowship by IIT Delhi. Parag Singla is supported by the DARPA Explainable Artificial Intelligence (XAI) Program with number N66001-17-2-4032. Both Mausam and Parag Singla are supported by the Visvesvaraya Young Faculty Fellowships by Govt. of India and IBM SUR awards. Any opinions, findings, conclusions or recommendations expressed in this paper are those of the authors and do not necessarily reflect the views or official policies, either expressed or implied, of the funding agencies.

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

APPENDIX

3 THEORY AND ALGORITHM

3.2 OBJECTIVE FUNCTION

**Lemma 1.** *The training loss $L(\Theta)$ as defined in eq. (1) is an inconsistent estimator of generalization error for 1oML, when $l_{\Theta}$ is a zero-one loss, i.e., $l_{\Theta}(\hat{\mathbf{y}}_{\mathbf{i}}, \mathbf{y}_{\mathbf{ij}}) = \mathbb{1}\{\hat{\mathbf{y}}_{\mathbf{i}} \neq \mathbf{y}_{\mathbf{ij}}\}$. (Proof in Appendix).*

*Proof.* Let $\mathcal{D}$ represent the distribution using which samples $(\mathbf{x}, \mathcal{Y}_{\mathbf{x}})$ are generated. In our setting, generalization error $\varepsilon(M_{\Theta})$ for a prediction network $M_{\Theta}$ can be written as: $\varepsilon(M_{\Theta}) = \mathbb{E}_{(\mathbf{x}, \mathcal{Y}_{\mathbf{x}}) \sim \mathcal{D}}(\mathbb{1}\{\hat{\mathbf{y}} \notin \mathcal{Y}_{\mathbf{x}}\})$, where $\hat{\mathbf{y}} = M_{\Theta}(\mathbf{x})$, *i.e.* the prediction of the network on unseen example sampled from the underlying data distribution. Assume a scenario when $\mathbf{Y}_{\mathbf{x}_{\mathbf{i}}} = \mathcal{Y}_{\mathbf{x}_{\mathbf{i}}}, \forall \mathbf{i}$, *i.e.*, for each input $\mathbf{x}_{\mathbf{i}}$ all the corresponding solutions are present in the training data. Then, an unbiased estimator $\hat{\varepsilon}_{\mathbb{D}}(M_{\Theta})$ of the generalization error, computed using the training data is written as: $\hat{\varepsilon}_{\mathbb{D}}(M_{\Theta}) = \frac{1}{m} \sum_{i=1}^{m} \mathbb{1}\{\hat{\mathbf{y}}_{\mathbf{i}} \notin \mathbf{Y}_{\mathbf{x}_{\mathbf{i}}}\}$. Clearly, the estimator obtained using $L(\Theta)$ (Naïve Objective), when the loss function $l_{\Theta}(\hat{\mathbf{y}}_{\mathbf{i}}, \mathbf{y}_{\mathbf{ij}})$ is replaced by a zero-one loss $\mathbb{1}\{\hat{\mathbf{y}}_{\mathbf{i}} \neq \mathbf{y}_{\mathbf{ij}}\}$, is not a consistent estimator for the generalization error. This can be easily seen by considering a case when $\hat{\mathbf{y}}_{\mathbf{i}} \in \mathbf{Y}_{\mathbf{x}_{\mathbf{i}}}$ and $|\mathbf{Y}_{\mathbf{x}_{\mathbf{i}}}| > 1$. $\qquad\square$

OPTIMIZATION ISSUES WITH CC-LOSS

For the task of PLL, Jin & Ghahramani (2002) propose a modification of the cross entropy loss to tackle multiplicity of labels in the training data. Instead of adding the log probabilities, it maximizes the log of total probability over the given target set. Inspired by Feng et al. (2020), we call it CC-LOSS:

$$L_{cc}(\Theta) = -\frac{1}{m} \sum_{\mathbf{i}=1}^{m} \log \left( \sum_{\mathbf{y}_{\mathbf{ij}} \in \mathbf{Y}_{\mathbf{x}_{\mathbf{i}}}} Pr\left(\mathbf{y}_{\mathbf{ij}} | \mathbf{x}_{\mathbf{i}}; \Theta\right) \right) \tag{9}$$

However, in the case of structured prediction, optimizing $L_{cc}$ suffers from numerical instability.

We illustrate this with an example. Consider solving 9 x 9 sudoku puzzle, $\mathbf{x}_{\mathbf{i}}$. The probabilty of a particular target board, $\mathbf{y}_{\mathbf{ij}}$, is a product of $r = 9^2 = 81$ individual probabilities over the discrete space $\mathcal{V} = \{1 \cdots 9\}$ of size 9, *i.e.*, $Pr(\mathbf{y}_{\mathbf{ij}} | \mathbf{x}_{\mathbf{i}}; \Theta) = \prod_{k=1}^{r} Pr(\mathbf{y}_{\mathbf{ij}}[k] | \mathbf{x}_{\mathbf{i}}; \Theta)$. In the beginning of the training process, the network outputs nearly uniform probability over $\mathcal{V}$ for each of the $r$ dimensions, making $Pr(\mathbf{y}_{\mathbf{ij}} | \mathbf{x}_{\mathbf{i}}; \Theta)$ very small $(= 9^{-81} \sim 5.09e{-}78)$. The derivative of $\log$ of such a small quantity becomes numerically unstable.

This issue is circumvented in the case of naïve loss by directly working with log probabilities and log-sum-exp trick [7]. However, in the case of CC-LOSS, we need to sum the probabilities over the target set $\mathbf{Y}_{\mathbf{x}_{\mathbf{i}}}$ before taking $\log$, and computing $Pr(\mathbf{y}_{\mathbf{ij}} | \mathbf{x}_{\mathbf{i}}; \Theta)$ makes it numerically unstable. Motivated by log-sum-exp trick, we use the following modifications which involves computing only log probabilities. For simplicity of notation, we will use $\Pr(\mathbf{y}_{\mathbf{ij}})$ to denote $\Pr(\mathbf{y}_{\mathbf{ij}} | \mathbf{x}_{\mathbf{i}}; \Theta)$ and $L_{cc}^{\mathbf{i}}$ to denote the CC Loss for the $\mathbf{i}^{th}$ training sample.

$$L_{cc}^{\mathbf{i}} = -\log \left( \sum_{\mathbf{y}_{\mathbf{ij}} \in \mathbf{Y}_{\mathbf{x}_{\mathbf{i}}}} Pr(\mathbf{y}_{\mathbf{ij}}) \right)$$

Multiply and divide by $maxp_{\mathbf{i}} = \max_{\mathbf{y}_{\mathbf{ij}} \in \mathbf{Y}_{\mathbf{x}_{\mathbf{i}}}} Pr(\mathbf{y}_{\mathbf{ij}})$:

$$L_{cc}^{\mathbf{i}} = -\log \left( maxp_{\mathbf{i}} \sum_{\mathbf{y}_{\mathbf{ij}} \in \mathbf{Y}_{\mathbf{x}_{\mathbf{i}}}} \frac{Pr(\mathbf{y}_{\mathbf{ij}})}{maxp_{\mathbf{i}}} \right)$$

---

[7]https://blog.feedly.com/tricks-of-the-trade-logsumexp/

Use the identity: $\alpha = \exp(\log(\alpha))$:

$$L_{cc}^{\mathbf{i}} = -\log(maxp_{\mathbf{i}}) - \log\left(\sum_{\mathbf{y_{ij}} \in \mathbf{Y_{x_i}}} \exp\left(\log\left(\frac{Pr(\mathbf{y_{ij}})}{maxp_{\mathbf{i}}}\right)\right)\right)$$

$$= -\log(maxp_{\mathbf{i}}) - \log\left(\sum_{\mathbf{y_{ij}} \in \mathbf{Y_{x_i}}} \exp\left(\log\left(Pr(\mathbf{y_{ij}})\right) - \log\left(maxp_{\mathbf{i}}\right)\right)\right)$$

In the above equations, we first separate out the max probability target (similar to log-sum-exp trick), and then exploit the observation that the ratio of (small) probabilities is more numerically stable than the individual (small) probabilities. Further, we compute this ratio using the difference of individual log probabilities.

**Lemma 2.** *Under the assumption $\mathbf{Y_{x_i}} = \mathcal{Y}_{\mathbf{x_i}}, \forall \mathbf{i}$, the loss $L'(\Theta) = \min_{\mathbf{w}} L_{\mathbf{w}}(\Theta, \mathbf{w})$, defined as the minimum value of $L_{\mathbf{w}}(\Theta, \mathbf{w})$ (defined in eq. (2)) with respect to $\mathbf{w}$, is a consistent estimator of generalization error for 1oML, when $l_{\Theta}$ is a zero-one loss, i.e., $l_{\Theta}(\hat{\mathbf{y}}_{\mathbf{i}}, \mathbf{y_{ij}}) = \mathbb{1}\{\hat{\mathbf{y}}_{\mathbf{i}} \neq \mathbf{y_{ij}}\}$.*

*Proof.* Let $\mathcal{D}$ represent the distribution using which samples $(\mathbf{x}, \mathcal{Y}_{\mathbf{x}})$ are generated. In our setting, generalization error $\varepsilon(M_{\Theta})$ for a prediction network $M_{\Theta}$ is:

$$\varepsilon(M_{\Theta}) = \mathbb{E}_{(\mathbf{x}, \mathcal{Y}_{\mathbf{x}}) \sim \mathcal{D}}(\mathbb{1}\{\hat{\mathbf{y}} \notin \mathcal{Y}_{\mathbf{x}}\})$$

where $\hat{\mathbf{y}} = M_{\Theta}(\mathbf{x})$, *i.e.* the prediction of the network on unseen example sampled from the underlying data distribution. Assume a scenario when $\mathbf{Y_{x_i}} = \mathcal{Y}_{\mathbf{x_i}}, \forall \mathbf{i}$, *i.e.*, for each input $\mathbf{x_i}$ all the corresponding solutions are present in the training data. Then, an unbiased estimator $\hat{\varepsilon}_{\mathbb{D}}(M_{\Theta})$ of the generalization error, computed using the training data is written as:

$$\hat{\varepsilon}_{\mathbb{D}}(M_{\Theta}) = \frac{1}{m}\sum_{i=1}^{m}\mathbb{1}\{\hat{\mathbf{y}}_{\mathbf{i}} \notin \mathbf{Y_{x_i}}\}$$

Now, consider the objective function

$$L'(\Theta) = \min_{\mathbf{w}} L_{\mathbf{w}}(\Theta, \mathbf{w}) = \min_{\mathbf{w}} \frac{1}{m}\sum_{\mathbf{i}=1}^{m}\sum_{\mathbf{y_{ij}} \in \mathbf{Y_{x_i}}} w_{\mathbf{ij}}\mathbb{1}\{\hat{\mathbf{y}}_{\mathbf{i}} \neq \mathbf{y_{ij}}\}$$

$$= \frac{1}{m}\sum_{\mathbf{i}=1}^{m}\min_{\mathbf{w_i}}\sum_{\mathbf{y_{ij}} \in \mathbf{Y_{x_i}}} w_{\mathbf{ij}}\mathbb{1}\{\hat{\mathbf{y}}_{\mathbf{i}} \neq \mathbf{y_{ij}}\}$$

$$\text{s.t. } w_{\mathbf{ij}} \in \{0,1\} \ \forall \mathbf{i}, \forall \mathbf{j} \text{ and } \sum_{\mathbf{j}=1}^{|\mathbf{Y_{x_i}}|} w_{\mathbf{ij}} = 1, \forall \mathbf{i} = 1\ldots m$$

For any $\mathbf{x_i}$, if the prediction $\hat{\mathbf{y}}_{\mathbf{i}}$ is correct, *i.e.*, $\exists \mathbf{y_{ij}}* \in \mathbf{Y_{x_i}}$ *s.t.* $\hat{\mathbf{y}}_{\mathbf{i}} = \mathbf{y_{ij}}*$, then $\mathbb{1}\{\hat{\mathbf{y}}_{\mathbf{i}} \neq \mathbf{y_{ij}}*\} = 0$ and $\mathbb{1}\{\hat{\mathbf{y}}_{\mathbf{i}} \neq \mathbf{y_{ij}}\} = 1, \forall \mathbf{y_{ij}} \in \mathbf{Y_{x_i}}, \mathbf{y_{ij}} \neq \mathbf{y_{ij}}*$. Now minimizing over $\mathbf{w_i}$ ensures $w_{\mathbf{ij}}* = 1$ and $w_{\mathbf{ij}} = 0 \ \forall \mathbf{y_{ij}} \in \mathbf{Y_{x_i}}, \mathbf{y_{ij}} \neq \mathbf{y_{ij}}*$. Thus, the contribution to the overall loss from this example $\mathbf{x_i}$ is zero. On the other hand if the prediction is incorrect then $\mathbb{1}\{\hat{\mathbf{y}}_{\mathbf{i}} \neq \mathbf{y_{ij}}\} = 1, \forall \mathbf{y_{ij}} \in \mathbf{Y_{x_i}}$, thus making the loss from this example to be 1 irrespective of the choice of $\mathbf{w_i}$. As a result, $L'(\Theta)$ is exactly equal to $\hat{\varepsilon}_{\mathbb{D}}(M_{\Theta})$ and hence it is a consistent estimator for generalization error. $\square$

### 3.3 GREEDY FORMULATION: MINLOSS

**Example 2.** *Consider a simple task with a one-dimensional continuous input space $\mathcal{X} \subset \mathcal{R}$, and target space $\mathcal{Y} = \{0, 1\}$. Consider learning with 10 examples, given as $(\mathbf{x} = 1, \mathbf{Y_x} = \{1\})$ (5 examples), $(\mathbf{x} = -1, \mathbf{Y_x} = \{0, 1\})$ (4 examples), $(\mathbf{x} = -2, \mathbf{Y_x} = \{1\})$ (1 example). The optimal decision hypothesis is given as: $\mathbf{y} = \mathbb{1}\{\mathbf{x} > \alpha\}$, for $\alpha \leq -2$, or $\mathbf{y} = \mathbb{1}\{\mathbf{x} < \beta\}$, for $\beta \geq 1$. Assume learning this with logistic regression using MINLOSS as the training algorithm optimizing*

*the objective in eq. (3). If we initialize the parameters of logistic such that the starting hypothesis is given by $\mathbf{y} = \mathbb{1}\{\mathbf{x} > 0\}$ (logistic parameters: $\theta_1 = 0.1$, $\theta_0 = 0$), MINLOSS will greedily pick the target $\mathbf{y} = 0$ for samples with $\mathbf{x} = -1$, repeatedly. This will result in the learning algorithm converging to the decision hypothesis $\mathbf{y} = \mathbb{1}\{\mathbf{x} > -0.55\}$, which is sub-optimal since the input with $\mathbf{x} = -2$ is incorrectly classified (fig. 1, see Appendix for a detailed discussion).*

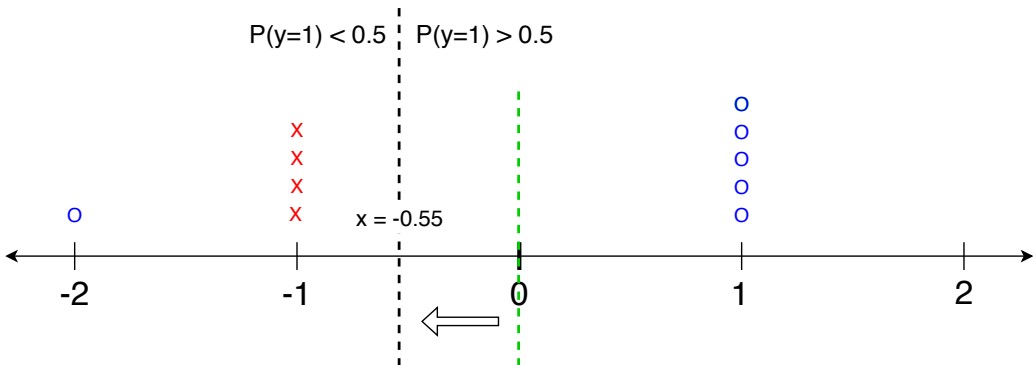

Figure 4: Decision Boundary learnt by logistic regression guided by MINLOSS. Green vertical line at $\mathbf{x} = 0$ is the initial decision boundary and black vertical line at $\mathbf{x} = -0.55$ is the decision boundary at convergence.

For logistic regression, when input $\mathbf{x}$ is one dimensional, probability of the prediction being 1 for any given point $\mathbf{x} = [x]$ is given as:

$$P(\mathbf{y} = 1) = \sigma(\theta_1 x + \theta_0) \quad \text{where} \quad \sigma(z) = \frac{1}{1 + e^{-z}}, z \in \mathcal{R}$$

The decision boundary is the hyperplane on which the probability of the two classes, 0 and 1, is same, *i.e.* the hyperplane corresponding to $P(\mathbf{y} = 0) = P(\mathbf{y} = 1) = 0.5$ or $\theta_1 x + \theta_0 = 0$.

Initially, $\theta_1 = 0.1$ and $\theta_0 = 0$ implies that decision boundary lies at $\mathbf{x} = 0$ (shown in green). All the points on the left of decision boundary are predicted to have 0 label while all the points on the right have 1 label. For all the dual label points ($\mathbf{x} = 1$), $P(\mathbf{y} = 1) < 0.5$, thus MINLOSS greedily picks the label 0 for all these points. This choice by MINLOSS doesn't change unless the decision boundary goes beyond -1.

However, we observe that with gradient descent using a sufficiently small learning rate, logistic regression converges at $\mathbf{x} = -0.55$ with MINLOSS never flipping its choice. Clearly, this decision boundary is sub-optimal since we can define a linear decision boundary ($\mathbf{y} = \mathbb{1}\{\mathbf{x} > \alpha\}$, for $\alpha \leq -2$, or $\mathbf{y} = \mathbb{1}\{\mathbf{x} < \beta\}$, for $\beta \geq 1$) that classifies all the points with label 1 and achieves 100% accuracy.

## 4 EXPERIMENTS

All the experiments are repeated thrice using different seeds. Hyperparameters are selected based on the held out validation set performance.

**Hardware Architecture:** Each experiment is run on a 12GB NVIDIA K40 GPU with 2880 CUDA cores and 4 cores of Intel E5-2680 V3 2.5GHz CPUs.

**Optimizer:** We use Adam as our optimizer in all our experiments. Initial learning rate is set to 0.005 for NLM (Dong et al., 2019) experiments while it is kept at 0.001 for RRN (Palm et al., 2018) experiments. Learning rate for RL phase is kept at 0.1 times the initial learning rate. We reduce learning rate by a factor of 0.2 whenever the performance on the dev set plateaus.

### 4.1 DETAILS FOR N-QUEENS EXPERIMENT

**Data Generation:** To generate the train data, we start with all possible valid 10–Queens board configurations. We then generate queries by randomly masking any 5 queens. We check for all

possible valid completions to generate potentially multiple solutions for any given query. Test data is also generated similarly. Training and testing on different board sizes ensures that no direct information leaks from the test dataset to the train dataset. Queries with multiple solutions have a small number of total solutions (2-6), hence we choose $\mathbf{Y_{x_i}} = \mathcal{Y}_{x_i}, \forall \mathbf{x_i}$.

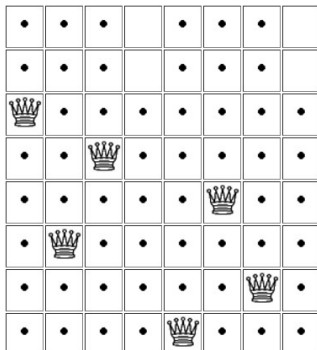 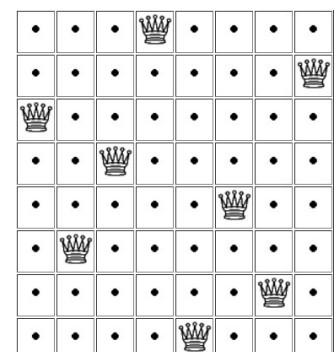 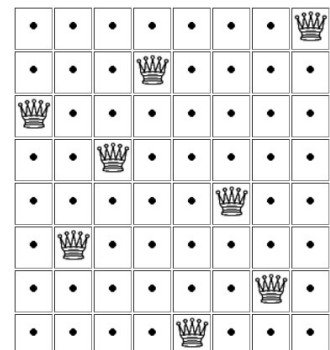

Figure 5: 8-Queens query along with its two possible solution.[8]

**Architecture Details for Prediction Network** $M_\Theta$**:** We use Neural Logic Machines (NLM)[9] (Dong et al., 2019) as the base prediction network for this task. NLM consists of a series of basic blocks, called 'Logic Modules', stacked on top of each other with residual connections. Number of blocks in an NLM architecture is referred to as its $depth$. Each block takes grounded predicates as input and learns to represent $M$ intermediate predicates as its output. See (Dong et al., 2019) for further details. We chose an architecture with $M = 8$ and $depth = 30$. We keep the maximum arity of intermediate predicates learnt by the network to be 2.

**Input Output for Prediction Network:** Input to NLM is provided in terms of grounded unary and binary predicates and the architecture learns to represent an unknown predicate in terms of the input predicates. Each cell on the board acts as an atomic variable over which predicates are defined.

**Unary Predicates:** To indicate the presence of a Queen on a cell in the input, we use a unary predicate, '*HasQueenPrior*'. It is represented as a Boolean tensor $\mathbf{x}$ of size $N^2$ with 1 on $k$ out of $N^2$ cells indicating the presence of a Queen. The output $\mathbf{y}$ of the network is also a unary predicate '*HasQueen*' which indicates the final position of the queens on board.

**Binary Predicates:** We use 4 binary predicates to indicate if two cells are in same row, same column, same diagonal or same off-diagonal. The binary predicates are a constant for all board configurations for a given size $N$ and hence can also be thought of as part of network architecture instead of input.

**Architecture Details for Selection Module** $S_\phi$**:** We use another NLM as our latent model $G_\phi$ within the selection module $S_\phi$. We fix $depth = 4$ and $M = 10$ for the latent model.

**Input Output for** $G_\phi$**:** Input to $G_\phi$ is provided in terms of grounded unary and binary predicates represented as tensors just like the prediction network. $G_\phi$ takes 1 unary predicate as input, represented as an $N^2$ sized vector, $\mathbf{y_{ij}} - \hat{\mathbf{y}}_{i\_}$, where $\hat{\mathbf{y}}_{i\_}$ is the prediction from its internal copy of the prediction network ($M_{\Theta\_}$) given the query $\mathbf{x_i}$. For each $\mathbf{y_{ij}} \in \mathbf{Y_{x_i}}$, $G_\phi$ returns a score which is converted into a probability distribution $Pr_\phi(\mathbf{y_{ij}})$ over $\mathbf{Y_{x_i}}$ using a softmax layer.

**Hyperparameters:**

The list below enumerates the various hyper-parameters with a brief description (whenever required) and the set of its values that we experiment with. Best value of a hyper-parameter is selected based on performance on a held out validation set.

1. **Data Sampling:** Since number of queries with multiple solutions is underrepresented in the training data, we up-sample them and experiment with different ratios of multi-solution

---

[8]Image Source: Game play on `http://www.brainmetrix.com/8-queens/`
[9]Code taken from: `https://github.com/google/neural-logic-machines`

queries in the training data. Specifically, we experiment with the ratios of $0.5$ and $0.25$ in addition to the two extremes of selecting queries with only unique or only multiple solutions. Different data sampling may be used during pre-training and RL fine tuning phases.

2. **Batch Size:** We use a batch size of 4. We selected the maximum batch size that can be accommodated in 12GB GPU memory.

3. *copyitr*: We experiment with two extremes of copying the prediction network after every update and copying after every 2500 updates.

4. **Weight Decay in Optimizer:** We experiment with different weight decay factors of 1E-4, 1E-5 and 0.

5. **Pretraining** $\phi$**:** We pretrain $G_\phi$ for 250 updates.

**Training Time:** Pre-training takes $10 - 12$ hours while RL fine-tuning take roughly $6 - 8$ hours using the hardware mentioned in the beginning of the section.

### 4.1 DETAILS FOR FUTOSHIKI EXPERIMENT

**Data Generation:** We start with generating all the possible ways in which we can fill a $N \times N$ grid such that no number appears twice in a row or column. For generating a query we sample any solution and randomly mask out $k$ positions on it. Also we enumerate all the $GreaterThan$ and $LessThan$ relations between adjacent pair of cells in the chosen solution and randomly add $q$ of these relations to the query. We check for all possible valid completions to generate potentially multiple solutions for any given query. Test data is also generated similarly. Training and testing on different board sizes ensures that no direct information leaks from the test dataset to the training data. Queries with multiple solutions have a small number of total solutions (2-6), so we choose $\mathbf{Y_{x_i}} = \mathcal{Y}_{\mathbf{x_i}}, \forall \mathbf{x_i}$ .

**Architecture Details for Prediction Network** $M_\Theta$**:** Same as N-Queens experiment.

**Input Output for Prediction Network:** Just like N-Queens experiment, the input to the network is a set of grounded unary and binary predicates. We define a grid cell along with the digit to be filled in it as an atomic variable. There are $N^2$ cells in the grid and each cell can take $N$ values, thus we have $N^3$ atomic variables over which the predicates are defined.

    **Unary Predicates:** To indicate the presence of a value in a cell in the input, we use a unary predicate, '*IsPresentPrior*'. It is represented as a Boolean tensor $\mathbf{x}$ of size $N^3$ with 1 on $k$ positions indicating the presence of a digit in a cell. The output $\mathbf{y}$ of the network is also a unary predicate '*IsPresent*' which indicates the final prediction of grid. Additionally, there are two more unary predicates which represent the inequality relations that need to be honoured. Since inequality relations are defined only between pair of adjacent cells we can represent them using unary predicates.

    **Binary Predicates:** We use 3 binary predicates to indicate if two vairables are in same row, same column, or same grid cell. The binary predicates are a constant for all board configurations for a given size $N$.

**Architecture Details for Selection Module** $S_\phi$**:** Same as N-Queens experiment.

**Input Output for** $G_\phi$**:** Same as N-Queens experiment except for the addition of two more unary predicates corresponding to the inequality relations. First unary predicate is $\mathbf{y_{ij}} - \hat{\mathbf{y}}_{\mathbf{i}\_}$ which is augmented with the inequality predicates.

**Hyperparameters:** Same as N-Queens experiment.

**Training Time:** Pre-training takes roughly $12 - 14$ hours while RL fine-tuning takes $7 - 8$ hours.

### 4.1 DETAILS FOR SUDOKU EXPERIMENT

#### DATA GENERATION FOR SUDOKU

We start with the dataset proposed by Palm et al. (2018). It has $180k$ queries with only unique solution and the number of givens are uniformly distributed in the range from 17 to 34. [10]. For the

---

[10]Available at `https://data.dgl.ai/dataset/sudoku-hard.zip`

queries with unique solution, we randomly sample 10000 queries from their dataset, keeping their train, val and test splits. Using the queries with 17-givens from the entire dataset of size $180k$, we use the following procedure to create queries with multiple solutions:

We know that for a Sudoku puzzle to have a unique solution it must have 17 or more givens (McGuire et al., 2012). So we begin with the set of 17-givens puzzles having a unique solution and randomly remove 1 of the givens, giving us a 16-givens puzzle which necessarily has more than 1 correct solution. We then randomly add 1 to 18 of the digits back from the solution of the original puzzle, while ensuring that the query continues to have more than 1 solution. [11] This procedure gives us multi-solution queries with givens in the range of 17 to 34, just as the original dataset of puzzles with only unique solution. We also observed that often there are queries which have a very large number of solutions ($> 100$). We found that such Sudoku queries are often too poorly defined to be of any interest. So we filter out all queries having more than 50 solutions. To have the same uniform distribution of number of givens as in the original dataset of puzzles with unique solution, we sample queries from this set of puzzles with multiple solutions such that we have a uniform distribution of number of givens in our dataset.

We repeat this procedure to generate our validation and test data by starting from validation and test datasets from Palm et al. (2018).

**Architecture Details for Prediction Network $M_\Theta$:** We use Recurrent Relational Network (RRN) (Palm et al., 2018) [12] as the prediction network for this task. RRN uses a message passing based inference algorithm on graph objects. We use the same architecture as used by Palm et al. (2018) for their Sudoku experiments. Each cell in grid is represented as a node in the graph. All the cells in the same row, column and box are connected in the graph. Each inference involves 32 steps of message passing between the nodes in the graph and the model outputs a prediction at each step.

**Input Output for Prediction Network:** Input to the prediction network is represented as a $81 \times 10$ matrix with each of the 81 cell represented as a one-hot vector representing the digits (0-9, 0 if not given). Output of the prediction network is a $81 \times 10 \times 32$ tensor formed by concatenating the prediction of network at each of the 32 steps of message passing. The prediction at the last step is used for computing accuracy.

**Architecture Details for Selection Module $S_\phi$:** We use a CNN as the latent model $G_\phi$. The network consists of four convolutional layers followed by a fully connected layer. The four layers have 100, 64, 32 and 32 filters respectively. Each filter has a size of $3 \times 3$ with stride of length 1.

**Input Output for $G_\phi$:** Similar to the other two experiments, the input to $G_\phi$ is the output $\hat{\mathbf{y}}_{\mathbf{i\_}}$ from the selection module's internal copy $M_{\Theta\_}$ along with $\mathbf{y_{ij}}$. Since the prediction network gives an output at each step of message passing, we modify the $G_\phi$ and the rewards for $S_\phi$ accordingly to be computed from prediction at each step instead of relying only on the final prediction.

**Hyperparameters:**

1. **Data Sampling:** Since number of queries with multiple solutions and queries with unique solution are in equal proportion, we no longer need to upsample multi-solution queries.

2. **Batch Size:** We use a batch size of 32 for training the baselines, while for RL based training we use a batch size of 16.

3. *copyitr*: We experiment with *copyitr* $= 1$ i.e. copying $M_\Theta$ to $M_{\Theta\_}$ after every update.

4. **Weight Decay in Optimizer:** We experiment with weight decay factor of 1E-4 (same as Palm et al. (2018)).

5. **Pretraining** $\phi$**:** We pretrain $G_\phi$ for 1250 updates, equivalent to one pass over the train data.

**Comparison with pretrained SOTA Model:** We also evaluate the performance of a pretrained state-of-the-art neural Sudoku solver (Palm et al., 2018)[13] on our dataset. This model trains and tests on instances with single solution. The training set used by this model is a super-set of the

---

[11]We identify all solutions to a puzzle using `http://www.enjoysudoku.com/JSolve12.zip`
[12]Code taken from: `https://github.com/dmlc/dgl/tree/master/examples/pytorch/rrn`
[13]Available at: `https://data.dgl.ai/models/rrn-sudoku.pkl`

Table 3: Mean test accuracy (±standard error) over three runs for multiplicity aware methods compared with baselines. *OS*: test queries with only one solution, *MS*: queries with more than one solution.

| | | *Naïve* | *Random* | *Unique* | **CC-Loss** | **MinLoss** | **I-Explr** | **SelectR** |
|---|---|---|---|---|---|---|---|---|
| **N-Queens** | *OS* | 70.59 ± 0.09 | 75.09 ± 0.33 | 72.91 ± 0.65 | 75.31 ± 0.45 | 77.29 ± 0.38 | 77.35 ± 1.07 | **79.73 ± 0.34** |
| | *MS* | 55.34 ± 2.82 | 66.85 ± 2.46 | 61.13 ± 1.13 | 75.76 ± 1.60 | 77.22 ± 1.28 | 79.46 ± 3.31 | **79.68 ± 1.35** |
| | **Overall** | 68.04 ± 0.46 | 73.72 ± 0.59 | 70.94 ± 0.71 | 75.39 ± 0.47 | 77.28 ± 0.48 | 77.70 ± 1.40 | **79.72 ± 0.46** |
| **Futoshiki** | *OS* | 65.59 ± 0.62 | 67.63 ± 0.96 | 65.49 ± 0.28 | 77.68 ± 0.34 | 76.78 ± 0.81 | **78.15 ± 0.65** | 78.01 ± 0.70 |
| | *MS* | 14.99 ± 2.17 | 19.13 ± 3.14 | 14.22 ± 2.77 | 69.30 ± 1.76 | 70.35 ± 1.16 | 70.88 ± 1.48 | **71.57 ± 1.02** |
| | **Overall** | 52.96 ± 0.96 | 55.53 ± 1.44 | 52.70 ± 0.74 | 75.59 ± 0.46 | 75.18 ± 0.64 | 76.33 ± 0.65 | **76.4 ± 0.36** |
| **Sudoku** | *OS* | 87.85 ± 0.84 | 89.19 ± 1.12 | **87.53 ± 0.82** | 88.26 ± 0.88 | 88.25 ± 0.35 | 88.73 ± 0.68 | 88.69 ± 0.55 |
| | *MS* | 09.13 ± 0.89 | 66.39 ± 2.82 | 13.65 ± 1.79 | 76.58 ± 1.63 | 76.93 ± 1.50 | 80.19 ± 1.51 | **81.73 ± 2.00** |
| | **Overall** | 48.49 ± 0.86 | 77.79 ± 1.96 | 50.59 ± 0.49 | 82.42 ± 0.45 | 82.59 ± 0.62 | 84.46 ± 0.69 | **85.21 ± 0.76** |

unique solution queries in our training data and contains 180,000 queries. This model achieves a high accuracy of 94.32% on queries having unique solution (*OS*) in our test data which is a random sample from their test data only, but the accuracy drop to 24.48% when tested on subset of our test data having only queries that have multiple solutions (*MS*). We notice that the performance on *MS* is worse than *Unique* baseline, even though both are trained using queries with only unique solution. This is because the pretrained model overfits on the the queries with unique solution whereas the *Unique* baseline early stops based on performance on a dev set having queries with multiple solutions as well, hence avoiding overfitting on unique solution queries.

**Training Time:** Pre-training the RRN takes around $20 - 22$ hours whereas RL fine-tuning starting with the pretrained model takes around $10 - 12$ hours.

## 4.3 Results and Discussions

Table 3 reports the mean test accuracy along with the standard error over three runs for different baselines and our three approaches. Note that the standard errors reported here are over variations in the choice of different random seeds and it is difficult to do a large number of such experiments (with varying seeds) due to high computational complexity. Below, we compare the performance gains for each of the seed separately.

**Seed-wise Comparison for Gains of SelectR over MinLoss**

In Table 4 we see that SelectR performs better than MinLoss for each of the three random seeds independently in all the experiments. We note that starting with the same seed in our implementation leads to identical initialization of the prediction network parameters.

Table 4: Seed wise gains of SelectR over MinLoss across different random seeds and experiments

| **Seed** | **Sudoku** | **NQueens** | **Futoshiki** |
|---|---|---|---|
| **42** | 3.12% | 3.40% | 0.69% |
| **1729** | 2.75% | 2.53% | 1.21% |
| **3120** | 1.98% | 1.39% | 1.77% |
| **Avg. Gain** | 2.61% | 2.44% | 1.22% |

**Details of the Analysis Depicted in Figure 3**

The large variation in the size of solution set ($|\mathcal{Y}_\mathbf{x}|$) in Sudoku allows us to assess its effect on the overall performance. To do so, we divide the test data into different bins based on the number of possible solutions for each test input ($\mathbf{x_i}$) and compare the performance of the best model obtained in the three settings: *Unique*, MinLoss and SelectR.

By construction, the number of test points with a unique solution is equal to the total number of test points with more than one solution. Further, while creating the puzzles with more than one solution, we ensured uniform distribution of number of filled cells from 17 to 34, as is done in (Palm et al., 2018) for creating puzzles with unique solutions in their paper. Hence, the number of points across different bins (representing solution count) may not be the same. Figure 6 shows the average size of each bin and the average number of filled cells for multiple solution queries in a bin. As we move to the right in graph (i.e., increase the number of solutions for a given problem), the number of filled cells in the corresponding Sudoku puzzles decreases, re-

Figure 6: #givens and #datapoints vs size of query's solution set

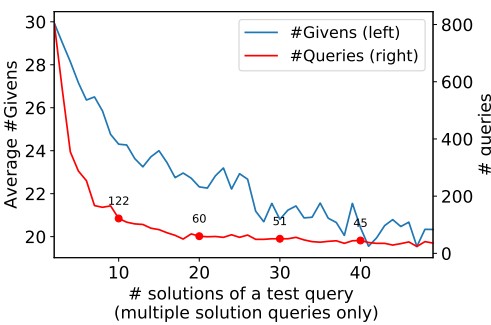

sulting in harder problems. This is also demonstrated by the corresponding decrease in performance of all the models in Figure 3. SELECTR is most robust to this decrease in performance.

**Discussion on Why SELECTR is better than I-EXPLR?**

In this section, we argue why SELECTR is more powerful than I-EXPLR, even though the reward structure for training the RL agent is such that eventually the $G_\phi$ in the RL agent will learn to pick the target closest to the current prediction (to maximize reward), and hence $S_\phi$ will be reduced to I-EXPLR.

We see two reasons why SELECTR is better than I-EXPLR.

First, recall that the I-EXPLR strategy gives the model an exploration probability based on its current prediction. But note that this is "only one" of the possible exploration strategies. For example, another strategy could be to explore based on a fixed epsilon probability. There could be several other such possible exploration strategies that could be equally justified. Instead of hard coding them, as done for I-EXPLR, our $G_\phi$ network gives the ability to learn the best exploration strategy, which may depend in complex ways on the global reward landscape (*i.e.*, simultaneously optimizing reward over all the training examples). Hence we use a neural module for this.

Second, note that I-EXPLR is parameter-free and fully dependent on $M_\Theta$, thus, has limited representational power of its own to explore targets. This is not the case with $G_\phi$. Its output $\bar{\mathbf{y}}$ and and the target ($\mathbf{y}_c$) closest to $M_\Theta$ prediction $\hat{\mathbf{y}}$ may differ i.e. $\bar{\mathbf{y}} \neq \mathbf{y}_c$ (see next paragraph for an experiment on this). When this happens, the gradients will encourage change in $\Theta$ so that $\hat{\mathbf{y}}$ moves towards $\bar{\mathbf{y}}$, and simultaneously encourage change in $\phi$ so that $\bar{\mathbf{y}}$ moves towards $\mathbf{y}_c$. That is, a stable alignment between the two models could be either of the two, $\mathbf{y}_c$ or $\bar{\mathbf{y}}$. This, we believe, increases the overall exploration of the model. Which of $\mathbf{y}_c$ or $\bar{\mathbf{y}}$ get chosen depends on how strongly the global landscape (other data points) encourage one versus the other. Such flexibility is not available to I-EXPLR where only $\Theta$ parameters are updated. We believe that this flexibility to explore more could enable SELECTR to jump off early local optima, thus achieving better performance compared to I-EXPLR.

We provide preliminary experimental evidence that supports that SELECTR explores more. For every training data point $q$, we check if the $\arg\max$ of $G_\phi$ probability distribution (i.e., highest probability $\bar{\mathbf{y}}$) and $\mathbf{y}_c$ differ from each other. We name such data points "exploratory". We analyze the fraction of exploratory data points as a function of training batches. See fig. 7. We observe that in the initial several batches, SELECTR has $3-10\%$ of training data exploratory. This number is, by definition, $0\%$ for I-EXPLR since it chooses $\bar{\mathbf{y}}$ based on model probabilities. This experiment suggests that SELECTR may indeed explore more early on.

Figure 7: Fraction of training samples for which $\arg\max$ of $G_\phi$ probability distribution is different from the target closest to model prediction. For I-EXPLR, this fraction is $0\%$

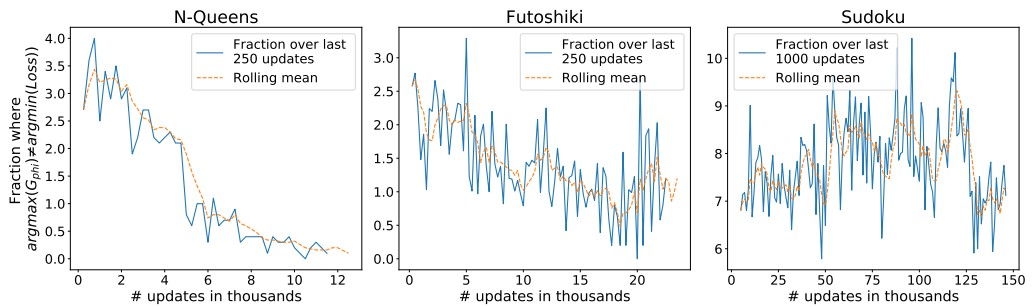

