# OpenReview forum: "Neural Learning of One-of-Many Solutions for Combinatorial Problems in Structured Output Spaces"
_ICLR.cc/2021/Conference — ICLR 2021 Poster_

### Official Review · AnonReviewer4 · 2020-10-27
**Paper defines interesting problem but motivation for RL based method could be improved**

**Rating:** 5
**Confidence:** 3

**Review:**


	1. Summary & contributions
This paper formalizes the 'one-of-many-learning' problem and proposes a method for solving this problem.

The paper first defines the one-of-many-learning problem, where a model must learn to map an input x to one of many possible targets y, where all possible y or only a subset may be given, in particular in a combinatorial setting. The paper explains failures of naïve approaches such as summing the loss over all possible y or only taking into account the y which has the lowest loss for the current model.

The paper then introduces the 'SelectR' framework where a separate neural network model (the selection module) is trained to predict which of the targets should be used for training the main model. The models are trained jointly using RL, where the selection module is trained to match the prediction of the main model and the main model minimizes the loss on the target selected by the selection module.

The authors demonstrate improved results compared to different (relatively simple) baselines in three different constraint satisfaction problems: N-Queens, Futoshiki and Sudoku.

	2. Strengths & weaknesses
This paper addresses an interesting problem which is well motivated, well defined and clearly and extensively related to other problem settings from the literature. The paper explains the problem clearly and gives good examples of failure modes of naïve approaches, motivating an alternate approach.

The paper shows that indeed performance improvements can be achieved by using such an alternate approach which explicitly considers the one-of-many-learning problem, by training a selection module to select the target used for training dynamically, in a joint fashion using RL.

The main weakness of the paper is that the specific proposed solution/framework (training a selection module using RL) is in my opinion not well motivated. The motivating problem (with the naïve MinLoss strategy) seems to be (lack of) exploration: can't this be addressed simply by adding some randomness (e.g. sampling target proportional to loss or model probability). Why is a separate neural network module needed?

Also, I do not understand the reward structure (# predicted variables equal to main model). It seems that the selection module is trained to basically match the prediction of the main model, and the paper states that this is 'since we do not know a-priori which y is optimal for defining the loss'. How does training a separate model to match the main model prediction help overcome this problem?

While the results do show the benefit of the 'SelectR' framework, I would like to see them compared to a simple (informed) strategy such as sampling a target according to model probability / loss / MinLoss with some epsilon probability of using a random target. The results may help in answering above questions.

Although I like the writing in general, I think the paper uses too much math notation and formalism. For example, the concepts of MinLoss and SelectR are relatively simple, but the notation in terms of one-hot w_ij makes things unnecessarily hard(er) to read and complicated. I think also the Lemma's and examples would be better explained with words than with heavy math, to help understanding. As a bonus, removing a lot of the math would allow for very helpful Algorithm 1 and maybe Figure 3 of the Appendix to be included in the main text.

	3. Recommendation
My current assessment is that the paper is marginally below the acceptance threshold.

	4. Arguments for recommendation
The paper addresses a well motivated and well explained problem and the results obtained using SelectR improve over the baselines, but it does not convince (enough) that such a (relatively complicated) approach is actually beneficial from a practical point of view as simpler alternatives are underexplored (see strengths & weaknesses). Additionally, I think the paper should use less formal notation as that will make the paper easier to read without losing the content.

	5. Questions to authors
See also strengths & weaknesses
- Do I understand correctly (from the formula for the cross entropy loss function) that Example 1 assumes a model which predicts y_1 and y_2 independently? I can imagine an autoregressive (structured) model which has p(y_1 = 1) = 0.5 and p(y_2 = 1|y1=1) = 0 and p(y_2 = 1|y_1 = 0) = 1 so p(1,0)=p(0,1)=0.5, which is optimal. It seems to me that the problem arises because of the independence structure of the variables in the model combined with the loss function (i.e. summing the log-probs for all targets).
- Why does Lemma 1, which is posed as a formalization of the problem arising in Example 1, consider a zero-one loss whereas Example 1 is based on a cross-entropy loss? This is confusing to me.
- The paper repeatedly mentions a 'prediction y^hat' as output of the main model. How is this defined? The model outputs a (structured) probability distribution but y^hat seems a vector. Is this vector a sample/argmax solution?


	6. Additional feedback

Minor comments/suggestions:
- The paper claims that compared to Neural Program Synthesis (NPS), where a generated program can be verified, in the setting considered in the paper there is no such additional signal available. However, the experiments all consist of problems where a solution can be  verified easily, even if it is outside the target set, so this does seem similar to NPS.
- It seems that forcing all probability mass to concentrate on one target can be helpful for some models (i.e. Example 1 if we assume independence of the variables), but may also be (unnecessarily) restrictive for other models which could more easily divide the probability mass. Maybe this would be interesting to discuss/investigate.
- The paper notes that 'one could also backpropagate the gradient of the expected loss given Pr(y_ij)'. This seems preferred to sampling, so a bit more discussion on why this was or was not used would be interesting.
- I would not consider the parameters of the main model as 'input' for the selection module. I would just say it takes as input the prediction from the main model.

---

> ### Author Response · Authors · 2020-11-16
> **Response to AnonReviewer4: Part 1**
>
> Thank you for the detailed review. Below, we have tried to address all of your concerns. Please let us know if we have left out anything, or something needs additional clarification.
>
> **Regarding a separate module for exploration and alternative of exploring based on model probability**
>
>
> We agree with the reviewer that the primary issue with MinLoss based approach is lack of exploration. And a simple strategy to incorporate exploration could be either based on using the probabilities of each of the possible target labels ($y$’s) or use an epsilon greedy RL approach. Such a simple strategy, though partially effective, may not extract the full information from the underlying problem structure and the associated solution landscape. Particularly, such a strategy would be local in nature, since it makes an independent decision for each data point. On the other hand, we argue that optimal exploration strategy may depend in complex ways on all the data points, viewed collectively, and could turn out to be significantly more complex, depending on the nature of the problem. In order to validate this thesis experimentally, based on the reviewer’s suggestion, we experimented with a simpler strategy that decides exploration based on the probability assigned by the prediction network to the targets in the set $Y_x$, referred to as I-ExplR (Informed Exploration Strategy). We find that, in line with our thesis, I-ExplR performs better than MinLoss on all the datasets. Further, the performance of I-ExplR is worse than our strategy of using a neural network ($G_\phi$) for exploration.
>
> As a meta-point, we would like to stress that, in our work, SelectR refers to the strategy of selecting a training label for each input, using an RL based framework. I-ExplR can be seen as a degenerate RL strategy which has an in-built exploration strategy but no learning. We have added relevant details in the paper including additional experiments. Besides contributing SelectR,  our paper’s contributions include defining the novel ML setting, and providing a solution framework for the problem of solution multiplicity.  Also, we will acknowledge the anonymous reviewer for suggesting I-ExplR.
>
>
> **Reward structure**
>
> Our reward structure is straightforward: for a given true label $\bar{y}$, which is one of the possible solutions for input $x$, the reward for selecting this label for further training the model is the Hamming distance of this label from the prediction $\hat{y}$, given the current model. Therefore, labels which are closer to the current prediction of the model will incur more reward, which makes intuitive sense.
> Note that If we were given an oracle to tell us which $\bar{y}$ is most suited for training $M_\Theta$, we wouldn’t have to use this proxy of Hamming distance. Instead, we would have trained the selection module to match the oracle.
>  We will be happy to add more details to clarify this in the paper.
>
>
> **Math notation and formalism;  Algorithm 1 and figure 3 in the main paper**
>
> Thank you for your feedback. In the next revision, we will be happy to add more explanations and intuitions in plain language to explain the concepts introduced in the paper. If there are any specific places, where you would like to see this happen, it will be great to get this feedback. However, we feel that since we are defining a new problem, it is important to lay down the foundations with proper formalism so that anyone working on it in the future has access to crisp definitions and proofs along with a clear understanding of the underlying concepts. We agree regarding Algorithm 1 and Figure 3, that they add clarity to the paper: we plan to use the extra page to move them to the main paper from the appendix.
>
>
> **Autoregressive (structured) model and independence of predictions**
>
> You correctly point out that the issue of solution multiplicity primarily arises due to the non-autoregressive model, where variables in the structured output space are decoded simultaneously (and therefore independently) from the underlying embeddings. Notably, most of the current state-of-the-art neural models for solving combinatorial problems work with non autoregressive models, e.g. SATNET [1], NLM [2], RRN[3], CNN for solving Sudoku [4]. We believe this might be because of their high efficiency of training and inference, since they do not have to decode the solution sequentially. We know of one variant of autoregressive models for our task [4], which predicts one output label at a time, but for each such prediction, it solves the full non-autoregressive decoding task. So, solution multiplicity problem will likely be a factor there too. Examining the value of 1oML for auto-regressive models will be a direction for future work after strong auto-regressive models for this task get developed. We will clarify in the paper that, for now, the scope of this work is within non-autoregressive models.
>
> *continued..*

---

> > ### Author Response · Authors · 2020-11-16
> > **Response to AnnonReviewer4: Part 2**
> >
> > *continued from Part 1*
> >
> >
> > **Zero-one loss in Lemma 1 vs Cross-entropy loss in Example 1**
> >
> > In the standard ML literature, the generalization error for binary classification is defined in terms of the 0-1 loss and hence we use it in Lemma 1. On the other hand, it is standard in neural network based models to use cross entropy loss as a differentiable proxy for the generalization error (which is based on 0-1 loss in our case). Hence, we demonstrate the practical learning issue in Example 1 using cross-entropy loss.
> >
> >
> > **Definition of prediction  $\hat{y}$**
> >
> > You have correctly pointed out that the neural network learns a (conditional) probability distribution over the output space (given the input $x$). Throughout our work, we consider prediction $\hat{y}$ as an argmax over this distribution. We will add a line to clarify this in the next version.
> >
> >
> > **Comparison with NPS**
> >
> > It is easy to verify a solution in our experiments only if we assume access to the underlying constraints (or rules) of the CSP. But we assume no such access to the underlying constraints, since that would change the overall problem (e.g., traditional solvers could then be used instead of using a neural network). Our setting is similar to that used in the earlier literature, e.g., SATNET [1], RRN [3], where the goal is to implicitly learn the constraints (for future prediction) purely from the training data given in the form of input output pairs, using a neural model. Our setting is clearly pointed out in the last paragraph of the introduction. Without access to the rules, there is no way for us to verify a solution, unlike in the case of NPS.
> >
> >
> > **Forcing all probability mass to concentrate on one target**
> >
> > Thanks for the note! As the reviewer has pointed out, forcing all probability mass to concentrate on one target is a natural choice for non-autoregressive models, where representing the entire joint distribution becomes intractable. We agree that for other models which can represent the joint distribution (e.g., autoregressive models), it may be a possibility to divide the probability mass over more than one correct solution, and worth exploring in the future. But as pointed out earlier, autoregressive models have not been the models of choice for these kinds of problems in the existing literature, which have mostly focused on non-autoregressive modeling.
> >
> >
> > **Gradient of the expected loss vs gradient of the loss at the sampled $\bar{y}$.**
> >
> > Yes, it is true that both strategies (1) compute the loss based on samples (2) compute the exact expected value of the loss, are possible in our setting, since we have a relatively small space of possible solutions to choose from, and exact computation of expected value is possible. In our experiments, we have used the strategy based on computing the expected value of the loss since it is exact. We have not tried the variation using sampling, since it is only an approximation to the exact computation and is typically used when computing exact value is intractable due to the large size of the underlying space.
> >
> >
> > **Parameters of the main model as ‘input’ to the selection module.**
> >
> > Thanks for pointing this out, we will correct this in the next revision.
> >
> > *References*
> >
> > [1] Po-Wei Wang, Priya L. Donti, Bryan Wilder, and J. Zico Kolter. Satnet: Bridging deep learning andlogical reasoning using a differentiable satisfiability solver. ICML 2019.
> >
> > [2] Honghua Dong, Jiayuan Mao, Tian Lin, Chong Wang, Lihong Li, and Denny Zhou.  Neural Logic Machines, ICLR 2019.
> >
> > [3] Rasmus  Berg  Palm,  Ulrich  Paquet,  and  Ole  Winther. Recurrent  relational  networks. NeurIPS  2018.
> >
> > [4] Kyubyong Park, Can Convolutional Neural Networks Crack Sudoku Puzzles?, https://github.com/Kyubyong/sudoku

---

> > > ### Comment · AnonReviewer4 · 2020-11-21
> > > **Thanks for updates, still some questions and too much math formalism**
> > >
> > > Thanks for the extensive answers and making updates to the paper. Please see my responses and additional questions below.
> > >
> > > **Exploration baseline**
> > >
> > > I especially appreciate the addition of the experiments with the I-ExpIR strategy, which, as expected, are a stronger baseline than the MinLoss strategy. I am still curious however where the improved performance of SelectR compared to this strategy comes from (see comment below on reward structure).
> > >
> > > **Reward structure**
> > >
> > > I still don't understand the motivation behind the reward structure. If I understand correctly, the reward is the hamming distance of the selection module prediction to the main model prediction, so it seems the SelectR module achieves maximum reward, thus behaves optimally, when it predicts y equal to the prediction of the main model. Therefore, the main model itself is optimal w.r.t. the selection learning task. This would be the I-ExpIR strategy? Why does a selection model that is trained to output the prediction of the main model give better results? It is not explained WHY this reward "captures the quality of selecting the corresponding target for training the prediction network".
> > >
> > > **Math formalism**
> > >
> > > I think the concepts in the paper are rather simple, which is actually a good thing, but the explanation is unnecessarily formal and complicated. I think many of the concepts and ideas could be very well (if not best) described with little or no math, so I do not consider the math notation 'proper formalism' but rather decorative. For example, the one-hot 'parameters' w only add unnecessary complexity: it is much easier to read about a strategy that chooses/learns a target y_i* \in Y_x_i for each x_i. This is much more intuitive and makes the math much simpler. The MinLoss strategy can be simply explained as picking the target which gives the lowest loss: therefore I think Eq 4 is not needed at all but could at least be rewritten as y_i* = argmin_y l(\hat{y}_i, y).
> > >
> > > As another example, I think Example 1 does not need all the formalism and could be written along the lines of: consider a problem where a model must (learn to) predict the target (y_1, y_2) = (1, 0) or (0, 1), but not (0, 0) or (1, 1). When minimizing cross entropy for a (fully factorized) model, the optimal solution is p(y_1) = p(y_2) = 1/2 which fails to recover the solution. As a side note, I think this example really needs the independence assumption of the model.
> > >
> > > Note: these examples are just to clarify my initial feedback. I don't expect the authors to rewrite the paper although I do think it would improve the paper.
> > >
> > > **Zero-one vs cross-entropy loss**
> > >
> > > I understand the zero-one vs cross-entropy loss, but Lemma 1 reads "The training loss as defined in eq. (1) … when l_\Theta is a zero-one loss", which, since l_\Theta is used in eq. (1) which is the training loss, suggests that the zero-one loss refers to training loss here. This is (still) confusing to me.
> > >
> > > **Knowledge of constraints**
> > >
> > > Your answer clarifies it for me, but from the paper the assumption that the task was to be solved without knowledge of the constraints was unclear to me.
> > >
> > > **Scope non-autoregressive models**
> > >
> > > I don't see the scope limiting to non-autoregressive models clarified in the paper?
> > >
> > > **Sample vs. expected loss**
> > >
> > > The phrase "Instead of backpropagating the loss gradient at a sampled target \hat{y}_i , one could also backpropagate the gradient of the expected loss given the distribution Pr_\phi(y_{ij}).", as written in the paper, suggests that you are using the sampled loss, while your answer states that you use expected loss.
> > >
> > > **Minor comment**
> > >
> > > Bottom page 3, \hat{y}_i(\hat{y}). Shouldn't this be \hat{y}(x_i)?

---

> > > > ### Author Response · Authors · 2020-11-23
> > > > **Response to additional questions by AnonReviewer4**
> > > >
> > > > Dear Reviewer,
> > > >
> > > > Thank you for your insightful comments. The latest version of the paper incorporates most of your suggestions.
> > > > Below, we address your concerns.
> > > >
> > > > **Reward structure**
> > > >
> > > > We agree with you that the alignment between $G_\phi$ and $M_\Theta$ is the eventual goal of the SelectR model. And hence, in the limit of convergence, $G_\phi$ predictions and $M_\Theta$ predictions will align, and $G_\phi$ will be reduced to I-ExplR. However, we see two reasons why SelectR is better than I-ExplR.
> > > >
> > > > First, recall that the I-ExplR strategy gives the model an exploration probability based on its current prediction. But we note that this is “only one” of the possible exploration strategies. For example, another strategy as originally suggested by the reviewer, could be to explore based on a fixed epsilon probability. There could be several other such possible exploration strategies that could be equally justified. Instead of hard coding them, as done for I-ExplR, our $G_\phi$ network gives the ability to learn the best exploration strategy, which may depend in complex ways on the global reward landscape (i.e., simultaneously optimizing reward over all the training examples). Hence we use a neural module for this.
> > > >
> > > > Second, note that I-Explr is parameter-free and fully dependent on $M_\Theta$, thus, has limited representational power of its own to explore targets. This is not the case with $G_\phi$. Its output $\bar{y}$ and and the target ($y_c$) closest to $M_\Theta$'s prediction $\hat{y}$ may differ i.e. $\bar{y} \ne y_c$ (see next para for an experiment on this). When this happens, the gradients will encourage change in $\Theta$ so that $\hat{y}$ moves towards $\bar{y}$, and simultaneously encourage change in $\phi$ so that $\bar{y}$ moves towards $y_c$ (as reward would be maximum for $y_c$ at this point). That is, a stable alignment between the two models could be either of the two, $y_c$ or $\bar{y}$. This, we believe, increases the overall exploration of the model. Which of $y_c$ or $\bar{y}$ get chosen depends on how strongly the global landscape (other data points) encourage one versus the other. Such flexibility is not available to I-Explr where only $\Theta$ parameters are updated. We believe that this flexibility to explore more could enable SelectR to jump off early local optima, thus achieving better performance compared to I-ExplR.
> > > >
> > > > We provide preliminary experimental evidence that supports that SelectR explores more. For every training data point $x$, we check if the argmax of $G_\phi$ probability distribution (i.e., highest probability $\bar{y}$) and $y_c$ differ from each other. We name such data points “exploratory”. We analyze the fraction of exploratory data points as a function of training batches. We observe that in the initial several batches, SelectR has 3-10% of training data exploratory. This number is, by definition, 0% for I-ExplR since it chooses $\bar{y}$ based on model probabilities. This experiment suggests that SelectR may indeed explore more early on.
> > > >
> > > > As a final note, we would like to thank the reviewer for very insightful comments and questions. We would be happy to include these discussions (and experiments) in the final version/appendix, which will add value to our paper.
> > > >
> > > >
> > > > **Math formalism**
> > > >
> > > > We appreciate your candid feedback and useful suggestions. We will try our best to incorporate as many intuitions as possible. That indeed is the purpose of the different examples and illustrations in the paper.
> > > >
> > > >
> > > > **Zero-one vs cross-entropy loss**
> > > >
> > > > Note that eqn 1 holds for any generic loss function, $l_\Theta$. It takes the value of cross entropy loss in example 1 while it refers to 0-1 loss in the Theorem.
> > > >
> > > >
> > > > **Knowledge of constraints**
> > > >
> > > > We do mention it in the first paragraph of the Introduction: “*These can be thought of as Constraint Satisfaction problems (CSPs) where the underlying constraints are not explicitly available, and need to be learned from training data*”.
> > > >
> > > > For additional clarity, in the latest revision of the paper, we have also stated this in the related works section when comparing with NPS.
> > > >
> > > >
> > > > **Scope non-autoregressive models**
> > > >
> > > > Based on your suggestion, we have added a note while motivating the new objective function on page 4.
> > > >
> > > >
> > > > **Sample vs expected loss**
> > > >
> > > > We have added a line in section 3.4 to clarify this.
> > > >
> > > >
> > > > **Minor Comment**
> > > >
> > > > We do not mean to write $\hat{y}_i$ as a function. The statement implies that $\hat{y}_i$ is the prediction for input $x_i$ and $\hat{y}$ denotes the prediction for input $x$.

---

### Official Review · AnonReviewer3 · 2020-10-28
**Contributions are borderline from first review**

**Rating:** 5
**Confidence:** 3

**Review:**

The quality and clarity of the paper is good overall. In my opinion the presentation is clear, the goal of the work, and the proposed solutions are presented cleanly. Authors give a few examples of the issues raised by learning from many correct solutions, which is appreciated.

In terms of significance. I believe the theory is not surprising at all, it is straightforward to see that eq.(1) will not be a consistent loss function for generalization. Now, in Lemma 2, given the definition of eq.(2), it is also not surprising to see that it is a consistent estimator, and in fact, the proofs are rather trivial. Thus, from the theoretical viewpoint these issues undermine the paper.
From the practical viewpoint, authors show that their proposed method SelectRL is better than other baselines, and my main concern in the practicality of the algorithm is that I don't see a strong case where SelectRL is significantly better than MinLoss, at least not statistically. Thus, my question here is, can authors claim that SelectRL is better than MinLoss from the experiments? If so, the gain seems small, and computationally speaking training an RL agent to select a solution seems an overkill.

Another question is: in the Random baseline, was the solution being picked uniformly at random? If so, wouldn't a distribution that is concentrated around a solution be better? I would've liked to see this in the experiments. If authors can comment on this question would be appreciated.

---

> ### Author Response · Authors · 2020-11-16
> **Response to AnonReviewer3**
>
> Thank you for your crisp review. Below we make an attempt to assuage all your concerns. Please let us know if more clarification would help and we would be happy to provide it.
>
> **Theory is not surprising at all**
> We would like to note that in this work,  we are defining a new problem and one of the primary contributions of our work is in laying out the framework for dealing with solution multiplicity, formally introducing all the definitions, fundamental theorems and their proofs, even if some of them are natural and not surprising. Once the concepts are crisply and clearly defined, the proofs may appear trivial to expert readers. To the best of our knowledge, the problem of solution multiplicity has not been hitherto addressed in the literature. Our major contributions in this work can be summarized as follows: 1. Define a novel problem. 2. Propose an intuitive, yet strong baseline along with its underlying theory and mathematical foundation (MinLoss), and, 3. Propose an RL based method to overcome the shortcomings of the greedy baseline via exploration (SelectR). In addition, we have also presented an extensive set of experiments showing the efficacy of our proposed model over the baseline approaches which ignore solution multiplicity altogether.
>
> **Can authors claim that SelectRL is better than MinLoss from the experiments?**
> Based on our experiments, we do believe that our SelectR based approach performs better than MinLoss. Table 3 (appendix) gives detailed numbers along with the standard error. We would like to note that the standard errors reported here are over variations in the choice of different random seeds in the experiments. Since it is difficult to do a large number of such experiments (with varying seeds) due to high computational complexity, we have also reported the comparisons using the (max) model based on a validation set for each of the algorithms. In both the settings of (mean) and (max), our SelectR based approach does better than MinLoss.
>
> In fact, SelectR performs better than MinLoss for each of the three seeds independently in all the experiments. We note that starting with the same seed in our implementation means that the initialization of the prediction network is the same. We have added a table (Table 4) in the appendix for seed-wise comparison of the gains of SelectR over MinLoss.
>
> In addition, we would like to note that the standard error arising out of the variation in the test set, is very very small (of the order of less than 0.005), and improvement of SelectR over all other algorithms is statistically significant with p values (computed using McNemar’s test for comparing MinLoss and SelectR using best models selected based on devset) of 1e-16,  0.03, and 1.7e-18 for NQueens, Futoshiki and Sudoku respectively. We also refer to the discussion about using a simpler exploration strategy in response to AnonReviewer4; please see the comment on “*Regarding a separate module for exploration and alternative of exploring based on model probability*”.
>
>
> **Computationally speaking training an RL agent to select a solution seems an overkill.**
> Computationally, we note that training time for SelectR is about 1.5 times more than MinLoss on average. Inference time is identical in both cases since it is only a forward propagation through the trained $M_\Theta$ network.
>
>
> **In the Random baseline, was the solution being picked uniformly at random?**
> Yes, we did pick the solution to be trained on uniformly at random among the set of possible solutions for each input ($x$). Once this was picked, we trained the entire algorithm using this solution (for a given input ($x$)). Choosing a uniform distribution seemed like a natural choice for a baseline since we did not have any natural preference for one solution over the others at the start of the training. We could possibly pick the solution to train on based on a distribution centered around a solution, but it is not clear what solution to centre the distribution on, and what this distribution should be. A more sophisticated strategy would pick a solution based on current model parameters in every iteration, but that is exactly what MinLoss and SelectR do.

---

### Official Review · AnonReviewer1 · 2020-10-28
**Interesting novel task, some concerns to address in the evaluation.**

**Rating:** 6
**Confidence:** 3

**Review:**

Summary:

The authors work in the domain of applying neural networks to combinatorial problems with structured output space, such as sudoku and n-queens. They notice how models currently performing well at this task encounter difficulties when there are multiple possible solutions. They formalize the task of learning any of multiple given (and possibly quite different) labels and propose an RL based approach to solve that task. They show improvements over selected baselines.

Great:
* The discussion of why the task is different from other instances of multiple labels is well argued and clear.
* Numerical examples are very helpful in following the description of the algorithms.
* The evaluation setting is well thought of: utilizing the state of the art model and comparing it to reasonable baselines (one of which is indeed current SoTA). The experiments seem reproducible, given the detailed descriptions in the appendix.


Could be improved:
* Table 2 is perhaps misleading. Table 3 with the same results with standard deviation gives a less clear answer on whether SelectR is actually always better than MinLoss (careful with significant digits, the standard deviation can’t have more than one!).
* The experiment depicted in figure 2 isn’t discussed, it doesn’t report confidence intervals, the distribution of training samples is not discussed (are there significantly less training examples with 50+ solutions, that could justify the drop in performance? A sudoku with this many solutions is likely to be quite sparse, does that affect performance?).


In summary, the paper introduces and formalizes an interesting novel task in the context of combinatorial problems with structured output space. While aspects in the evaluation could be clarified, the paper is clear and interesting. I recommend an accept, and would be willing to increase the score if my concerns are addressed.

---

> ### Author Response · Authors · 2020-11-16
> **Response to AnonReviewer1**
>
> Thank you for your thoughtful review. Below we hope to address the concerns raised by you. Please let us know if there are still any doubts and we would be happy to address them.
>
> **Table 2 is perhaps misleading. Is SelectR actually better than MinLoss**
> Thanks for the careful read of the appendix. We believe your concern arises out of the seemingly “high” standard error numbers reported in Table 3. We would like to note here that the standard errors reported here are over variations in the choice of different random seeds in the experiments. Since it is difficult to do a large number of such experiments (with varying seeds) due to high computational complexity, we have also reported the comparisons using the (max) model based on a validation set for each of the algorithms. In both the settings of (mean) and (max), our SelectR based approach does better than MinLoss.
> In fact, SelectR performs better than MinLoss for each of the three seeds independently in all the experiments. We note that starting with the same seed in our implementation means that the initialization of the prediction network is the same. We have added a table (Table 4) in the appendix for seed-wise comparison of the gains of SelectR over MinLoss.
> In addition, we would like to note that the standard error arising out of the variation in the test set, is very very small (of the order of less than 0.005), and improvement of SelectR over all other algorithms is statistically significant with p values (computed using McNemar’s test for comparing MinLoss and SelectR using best models selected based on devset) of 1e-16,  0.03, and 1.7e-18 for NQueens, Futoshiki and Sudoku respectively.
> We have made this explicit in the newly uploaded version of the paper. If there are any additional concerns, please let us know and we will be happy to address them.
>
> **Significant digits in std dev**
> We have reported both mean and std error up to 2 decimal places. We are not sure why the number of significant digits in std error should be less than that in the mean. We would appreciate it if you could elaborate, or share a reference.
>
> **Discussion on experiment depicted in figure 2**
> First, we would like to point out that due to an oversight, the curves depicted in Figure 2 correspond to the models trained using a randomly picked seed value for each of the algorithms and not for the best model (max) chosen based on the validation set as reported in Table 2. We apologize for this mistake which has been corrected in the newly uploaded version of the paper. Trends remain the same in the updated graph, but the effects are more pronounced. Below, we give further details as requested in the review:
> 1. The trained models are fixed across all the points in the Figure for each of the algorithms. These models exactly correspond to those used for obtaining the results reported in Table 2 (max models selected using a validation set). The test data is also the same as used for computing Table 2 results. The only difference is that now we have divided the test data into different bins based on the number of possible solutions for each test input (x) for further analysis and insights.
> 2. By construction, the number of test points with a unique solution is equal to the total number of test points with more than one solution [Section 4.1#Sudoku]. Further, while creating the puzzles with more than one solution, we ensured uniform distribution of the number of filled cells from 17 to 34, as is done in Palm et al. 2018, for creating puzzles with unique solutions in their paper. Hence, the number of points across different bins (representing solution count) may not be the same in Figure 2. In the appendix, we have added a plot showing the average size of each bin and the average number of filled cells for queries in a bin.
> 3. As correctly pointed by the reviewer, as we move to the right in the graph (i.e., increase the number of solutions for a given problem), the number of filled cells in the corresponding Sudoku puzzles decreases, resulting in harder problems. This is also demonstrated by the corresponding decrease in the performance of all the models in Figure 2. SelectR is most robust to this decrease in performance.
> 4. We have added 95% confidence intervals for the points in Figure 2 in the newly uploaded version of the paper. All our differences are statistically significant (comparison is across max models for each of the algorithms).
>
> We have also added the above details in the main paper and appendix. We will be happy to answer additional questions if any.

---

### Official Review · AnonReviewer2 · 2020-10-28
**Convincing new learning approach for multiple-solution combinatorial problems**

**Rating:** 8
**Confidence:** 4

**Review:**

This paper aims at devising a targeted approach that takes into account the specific structure of combinatorial problems with multiple solutions. The proposed approach leverages RL to select the best targets among the solution sets at each iteration. SelectR convincingly outperforms both the naive and a cleverer baseline, showcasing the applicability of the method.

Originality
The problem of interest is relevant to a number of machine learning applications but has largely been ignored by the community up until now, as is made clear in the very complete related works section of the paper. Notably, the question of selecting the best target for learning is, as far as I am aware, novel. Consequently, so is both the approach and the baselines it is compared against.

Significance
As ML practitioners try their hand at more and more complex problems, this approach will become more and more relevant.
Further, since this paper is the first to define the one-of-many problem, sets out to define the general framework, and defines reasonable baseline, it is very relevant.
The effort made to link the problem of interest with existing other problems mean that it's easier for readers to draw parallels, and helps bolster the paper's significance. For instance, the experimental results tend to show that naively using multiple possible solutions is worse than ignoring these data points. This is in direct contradiction with the general consensus for tasks such as machine translation, where multi-solution datasets are not available, but are longed for.

Clarity
Overall, the paper is well-written and easy to follow. A couple of things might be made clearer, though, including:
- the description of the pretraining regimen, which is a bit convoluted. It would probably help to refer to the internal M for the selection module as a target network, which it seems to be.
- the description of the reward for the selection module is a bit complicated too, and the the fact that solutions can be split into r components could be reminded here.
It would also be helpful to give more insight into why this reward was chosen (I imagine this partial-reward makes it easier to 'see' some reward than a reward for exact matching, but I'm speculating here), and what the consequences of this choice are (aside from improved performance). Does the selection module opt for the 'easiest' targets to predict? Do the targets chance as training goes along? Could the selection module be trained at a meta-level rather than at the transition-level?


Overall, this is a nicely-written paper offering a novel approach to a significant problem, and showcasing its performance improvements. It would make a nice addition to this year's ICLR.

---

> ### Author Response · Authors · 2020-11-16
> **Response to AnonReviewer2**
>
> We thank the reviewer for a very encouraging feedback and review. Below we try to address the few concerns raised by the reviewer regarding clarity in some of the aspects.
>
> **Description of the pre-training regimen**
> Here are the two steps of pre-training: (a) $M_\Theta$ is pre-trained using either MinLoss on all the data or using MinLoss over data points with unique solutions, which is decided using a validation set. (b) $\Theta$ parameters learned in step (a) are copied into $M_{\Theta_-}$, the copy of the prediction network inside the selection module. Keeping  $\Theta$ and $M_{\Theta_-}$ fixed, $G_\phi$ is trained. In this sense, $G_\phi$ is being trained to mimic (maximize) the reward obtained using the prediction from the $M_\Theta$ network. If you meant to say that the copy of $M$ inside the selection module may be called ‘target network’ since $G_\phi$ is trying to mimic it, this may not be entirely correct. Since the reward to train $G_\phi$ is coming externally based on the similarity of prediction with the chosen target, i.e., based on Hamming distance between the two, which is not identical to the probabilities used in computing the cross-entropy loss. We will add more details to clarify this in the paper.
>
> **Description of the reward**
> The job of the selection module is to pick one target, $\bar{y}$, among the possible targets for each input ($x$) for training the prediction network $M_\Theta$. Intuitively, $\bar{y}$ would be a good choice for training $M_\Theta$, if it is “easier” for the model to predict $\bar{y}$. In our reward design, we measure this degree of ease using hamming distance between $\bar{y}$ and $M_\Theta$ prediction $\hat{y}$. Specifically, the reward associated with a certain choice of $\bar{y}$ is the similarity of the current prediction of the model, $\hat{y}$, with $\bar{y}$ where the similarity is measured in terms of the number of matching grid cells. This detail is described in the paragraph preceding Equation 6 in the paper. We will make it more prominent in the updated version of the paper to have additional clarity. We note that if we had an oracle to tell us which $\bar{y}$ to train on for a given input ($x$), to get the optimal model, the selection module can be trained using that oracle. Since we do not have such an oracle, a reward based scheme as described above is used.
> In general, the selection network would pick the “easiest” target for input ($x$) (with the notion of easiness as defined above). Since there is an exploration and exploitation trade-off in an RL setting, often it might pick those targets which are not “easy” (i.e., with greater hamming distance from the current model prediction) as a way of exploration. In order to examine the number of target switchings during training, we performed the experiment described below.
>
> **Do the targets change as training goes along?**
> Based on your question, we performed some additional experiments where we calculated the fraction of data points with more than one solution for which the MinLoss target (closest to the current prediction) switched as learning proceeds. We note that the change in the MinLoss based target is an important indicator since this would tell us that the closest point to current model prediction has moved, presumably due to RL based exploration. We observed an interesting trend for all the datasets: In the beginning, this fraction increases as the learning proceeds, indicating that a lot of switching of targets was happening. The switching fraction being small in the beginning can be explained by the fact that pre-trained model parameters ($\Theta$) have just started to move. This fraction increases as learning proceeds due to RL based exploration in SelectR and simultaneous changing of the parameters $\Theta$ of the prediction network $M$. Towards the end of training, this fraction again comes down showing that learning had stabilized; in our experiments, the value remained high for Sudoku, which we believe is due to the fact that we only had a subset of all possible solutions for training.
>
> **Training selection module at a meta-level**
> We did not quite understand what you mean by the comment “Could the selection module be trained at a meta-level rather than at the transition-level?” We request you to please elaborate, and we will be happy to respond.

---

### Author Response · Authors · 2020-11-16
**Common response to all reviewers**

Dear reviewers,

We thank you for such thought-provoking reviews. In response, we have updated the main paper and appendix with factual data (graphs, tables, significance tests etc.) and have addressed clarification doubts separately for each of the reviews below. In the next revision, we will add all the clarifications in the paper.

Thanks!

---

> ### Author Response · Authors · 2020-11-22
> **New revision of the paper**
>
> Dear reviewers,
>
> As promised, we have incorporated all the suggestions and clarifications in the latest revision. We acknowledge that there are some additional comments by *AnonReviewer4*. We will address them soon.
>
> Thanks!

---

### Author Response · Authors · 2020-11-25
**Summary of the major changes in the latest revision**

Dear reviewers,

We thank you once again for your insightful comments on our work. Below we provide a summary of the changes that are included in the latest revision of our paper. If you still feel that some aspects need more clarity or discussion, we would be happy to address them in the final revision.

Lastly, we notice that a few reviewers have addressed MinLoss as a baseline strategy. We would like to politely request that since 1oML is a new task, connecting it to MinLoss should also be seen as a contribution of our paper. Of course other tasks (like distant supervision) have considered similar loss functions, so it is definitely not a major contribution — however calling it a baseline presupposes the existence of such an approach for the task, which is not the case here.

Thanks!

**Summary of the changes:**

1. **Informed exploration strategy (I-ExplR)**: As suggested by *AnonReviewer4*, we experimented with an *informed* exploration strategy (I-ExplR) and compared it with our proposed *Learning* based exploration (SelectR): changes are in section 3.4 (1st para, pg 6), section 4.3 (Results & Discussion) and Table 2. We will be acknowledging the reviewer in the final version for suggesting it and would be happy to add more discussion in the appendix if required.

2. **Statistical significance of our gains:** explicitly mentioned the p-values for statistical tests (section 4.3, pg 9, below Table 2).

3. **Seed-wise comparison of the gains across tasks:** Added in the Appendix, Table 4 on pg 18.

4. **Analysis depicted in figure 3 (earlier figure 2):** updated the figure to add confidence intervals. Added a detailed description and analysis of the experiment in Appendix, pg 19. Also added a graph (figure 6, pg 19) to assist the discussion. Please note that there was a mistake in figure 2 of the original manuscript: it was not comparing the best models obtained by the different methods. We apologize for this mistake which has been corrected now. Trends remain the same in the updated graph, but the effects are more pronounced.

5. **Clarification and intuition of our reward structure:** section 3.4 ( under the description of *"Update of $\phi$ Parameters"*, above eqn 6 on pg 6).

6. Moved Algorithm 1 and illustrative figure for example 2 from the appendix to the main paper.

7. Clarification while comparing with NPS in related works (section 2, last line under ‘Solution Multiplicity in Other Settings’).

8. Clarification in the input / output specification of selection module:  section 3.4 (under the description of *"Selection Module $S_\phi$"*, pg 6)

9. Added a note on our focus on non-autoregressive models ( pg 2)

---

### Decision · Program_Chairs · 2021-01-07
**Final Decision**

**Decision:**

Accept (Poster)

**Comment:**

The reviewers are enthusiastic about this work, and the few comments that they had were appropriately addressed by the reviewers.